# Multi-omics analysis reveals contextual tumor suppressive and oncogenic gene modules within the acute hypoxic response

Zdenek Andrysik[1,2], Heather Bender[1,2], Matthew D. Galbraith [1,2✉] & Joaquin M. Espinosa [1,2,3✉]

Cellular adaptation to hypoxia is a hallmark of cancer, but the relative contribution of hypoxia-inducible factors (HIFs) versus other oxygen sensors to tumorigenesis is unclear. We employ a multi-omics pipeline including measurements of nascent RNA to characterize transcriptional changes upon acute hypoxia. We identify an immediate early transcriptional response that is strongly dependent on HIF1A and the kinase activity of its cofactor CDK8, includes indirect repression of MYC targets, and is highly conserved across cancer types. HIF1A drives this acute response via conserved high-occupancy enhancers. Genetic screen data indicates that, in normoxia, HIF1A displays strong cell-autonomous tumor suppressive effects through a gene module mediating mTOR inhibition. Conversely, in advanced malignancies, expression of a module of HIF1A targets involved in collagen remodeling is associated with poor prognosis across diverse cancer types. In this work, we provide a valuable resource for investigating context-dependent roles of HIF1A and its targets in cancer biology.

[1] Department of Pharmacology, School of Medicine, University of Colorado Anschutz Medical Campus, Aurora, CO, USA. [2] Linda Crnic Institute for Down Syndrome, School of Medicine, University of Colorado Anschutz Medical Campus, Aurora, CO, USA. [3] Department of Molecular, Cellular and Developmental Biology, University of Colorado Boulder, Boulder, CO, USA. ✉email: matthew.galbraith@cuanschutz.edu; joaquin.espinosa@cuanschutz.edu

Hypoxia plays an important role in both physiological and pathological processes in humans and is a common feature of solid tumors as a consequence of rapid proliferation and aberrant vascularization[1]. The hypoxia-inducible transcription factors HIF1A and HIF2A are well-known regulators of the response to hypoxia, inducing genes involved in metabolic reprogramming, angiogenesis, tissue remodeling, stemness, and immune regulation[1]. In cancer, HIFs have been described as oncogenes or tumor suppressors in different settings. HIF activity has been associated with increased mortality, metastasis, stemness, immune evasion, and resistance to therapy[1–4], thus providing rationale for therapeutic targeting of these transcription factors in cancer. Indeed, HIF2A-specific inhibitors have shown promise in treating clear cell renal cell carcinoma[2]. However, both HIF1A and HIF2A can also exert tumor-suppressive effects in certain contexts[5,6], and the mechanisms driving this functional duality are unclear. Therefore, a deeper understanding of gene expression programs driven by HIFs and their roles in cancer biology is necessary for better design and use of HIF-based interventions.

Although both HIF1A and HIF2A are stabilized upon exposure to hypoxia and regulate overlapping sets of target genes, evidence suggests that they may preferentially regulate short- versus long-term hypoxia, respectively[7]. In addition, rapid HIF-independent $O_2$-sensing mechanisms have recently been identified, raising the possibility that the earliest hypoxic signaling may not be driven by the HIFs[8–10]. To date however, studies of HIF-dependent and -independent transcriptional responses to hypoxia have focused on steady-state mRNA levels at relatively long time points[8–15], rather than directly measuring changes in transcription, preventing a clear dissection of HIF-driven direct versus indirect effects on the transcriptome.

In this work, in order to better define the early transcriptional response to acute hypoxia, we measured changes in transcription using Precision Run-On with sequencing (PRO-seq). PRO-seq only detects transcriptionally engaged RNA polymerases, thus enabling rapid and direct measurement of changes in nascent RNA synthesis[16]. Combined with measurement of HIF1A chromatin binding and steady-state mRNA levels in multiple cell lines, PRO-seq enabled a deep characterization of the HIF1A-driven transcriptional response to acute hypoxia. Our analysis of this comprehensive data set identified both protein-coding and non-coding HIF1A target genes as part of the acute response and produced several mechanistic insights. Hypoxia induces acute transactivation and repression in a HIF1A-dependent manner, but only transactivation is associated with HIF1A binding. Repressed genes are enriched for MYC targets, which could be explained by transactivation of MXI1, a HIF1A target known to attenuate MYC-driven transcription. HIF1A-dependent transactivation involves the release of paused RNA polymerase II (RNAPII) and requires the kinase activity of CDK8, a known HIF1A cofactor. Interestingly, a subset of HIF1A targets, including DDIT4, are sensitive to basal HIF1A levels in normoxia. The acute response is highly conserved across cancer cell types and is associated with strong HIF1A binding to promoters and actively transcribed enhancers. Analysis of genetic screen data reveals that, in normoxia, HIF1A behaves as a strong tumor suppressor, which is genetically linked to its target DDIT4, a repressor of mTOR signaling. Contrastingly, in human cancers, a subset of HIF1A targets involved in remodeling of the extracellular matrix is consistently associated with poor prognosis. Our data sets and analyses thus provide a resource for advanced understanding of the role of HIF1A and its targets in cancer biology.

## Results

### Identification of the acute transcriptional response to hypoxia.
In order to identify rapid changes in transcriptional activity upon hypoxia, we performed PRO-seq in HCT116 wild-type (WT) and HIF1A$^{-/-}$ cells under normoxia (21% $O_2$) and after 90 min of exposure to hypoxia (1% $O_2$) (Fig. 1a). This short treatment time was sufficient to induce both intracellular hypoxia, as detected in real-time using the fluorogenic hypoxia reporter Image-iT Red (Supplementary Fig. 1a), and stabilization of HIF1A and HIF2A (Fig. 1b). In addition, we measured HIF1A chromatin binding by ChIP-seq and changes in the steady-state transcriptome using poly(A) + RNA-seq after longer-term exposure to hypoxia (24 h) (Fig. 1a). This later time point was chosen to capture persistent changes in steady-state mRNA and because HIF1A chromatin binding patterns have been shown to be stable over time[14].

At most human protein-coding genes, shortly after initiation, elongating RNAPII pauses at ~20–60 nucleotides downstream of transcription start sites (TSS), due to the action of negative elongation factors and before recruitment of positive elongation factors enables further elongation[17,18]. This regulated rate-limiting step results in typical PRO-seq profiles with peaks of high read density near the TSS, representing paused RNAPII, and lower coverage throughout gene bodies, corresponding to elongating RNAPII. Thus, to identify changes in productive transcription in response to acute hypoxia, we first quantified PRO-seq signals within gene body regions and tested for differential transcriptional activity using DESeq2 (ref. [19]) on data from biological replicates (see Methods, Supplementary Data 1 and Supplementary Fig. 1b, c for replicate comparison). Rapid transcriptional activation and repression were apparent after 90 min of hypoxia in WT cells (Fig. 1c). This early transcriptional response was largely dependent on the presence of HIF1A, with both activation and repression reduced in HIF1A$^{-/-}$ cells (Fig. 1c, d), and included many well-characterized HIF1A targets such as ENO1, PDK1, HK1, and SLC2A1 (Supplementary Fig. 1d). Genome browser views for example downregulated and unchanged genes are shown in Supplementary Fig. 1e. Thus, while other $O_2$-sensitive factors such as lysine demethylases may rapidly modulate the chromatin environment[8,9], our data indicate that HIF1A plays a critical functional role in driving the acute transcriptional response to hypoxia. The residual changes observed in HIF1A$^{-/-}$ cells could be driven by HIF2A, which is expressed in HCT116 cells and stabilized upon hypoxia (Fig. 1b).

Gene set enrichment analysis (GSEA) of the acute transcriptional response indicates positive enrichment of known hypoxia- and glycolysis-related genes (Fig. 1e, Supplementary Fig. 1f), and the upstream regulator analysis module of Ingenuity Pathway Analysis (IPA) suite correctly predicts inactivation of the prolyl hydroxylases that repress HIF activity (EGLNs) concurrently with activation of HIF1A (Supplementary Fig. 1g). Interestingly, GSEA reveals early repression of MYC targets (Fig. 1e, Supplementary Fig. 1f), which could be explained by HIF1A-dependent induction of the MYC repressor MXI1 (refs. [20–23]), a gene that we found is indeed induced at the early time point by PRO-seq (Fig. 1c, d, Supplementary Fig. 1d, Supplementary Data 1). Importantly, a comparison of the acute transcriptional response identified by PRO-seq with 22 published studies characterizing the hypoxic response using more traditional approaches revealed that many genes in our data set have not previously been linked to hypoxic signaling (Supplementary Data 2), thus providing a resource to further characterize hypoxia-regulated pathways.

In many transcriptional networks, stimulus-responsive genes are associated with the release of pre-loaded, paused RNAPII[17]. Indeed, relative to their gene bodies, acute hypoxia-inducible genes display relatively modest changes in transcriptional activity at TSSs and consequent decreases in pausing index (PI), with an impaired response in HIF1A$^{-/-}$ cells (Fig. 1f, g). By contrast, acute hypoxia-repressed genes (defined by decreases in gene body

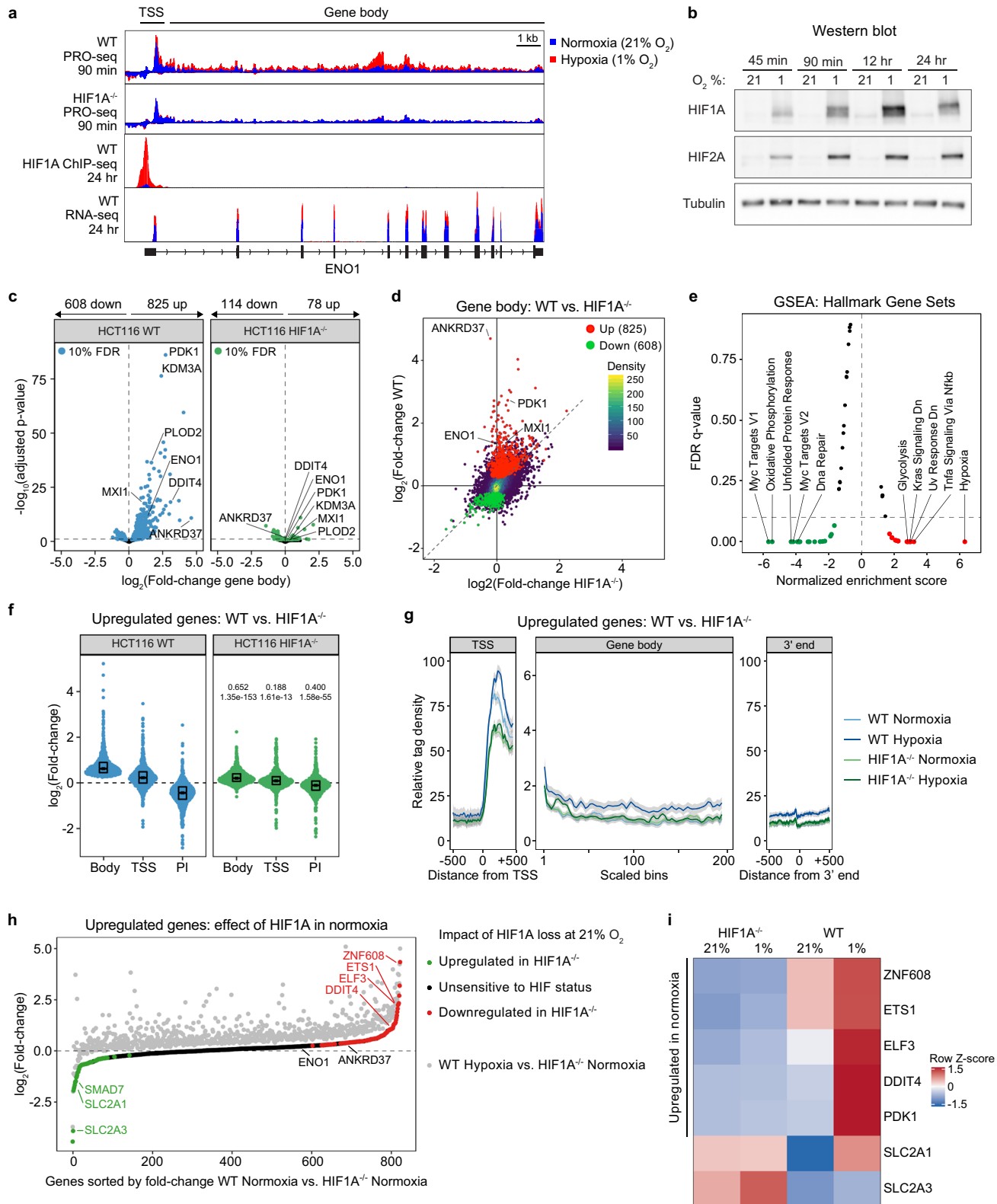

activity) display mildly decreased signals at TSS along with increased PI values (Supplementary Fig. 2a–c). These observations are consistent with the ability of HIF1A to induce recruitment of transcription elongation factors to sites of active chromatin[15,24,25].

Notably, depletion of HIF1A does not have an obvious global impact on transcription during normoxia or hypoxia, with WT and HIF1A$^{-/-}$ cells displaying highly correlated transcriptional activity at both TSSs and gene bodies (Supplementary Fig. 2d, e). However, the metagene profiles of HIF1A$^{-/-}$ cells show mildly decreased signals at TSS (Fig. 1g, Supplementary Fig. 2b, c). Using DESeq2 analysis, we identified 386 TSS regions and 1979 gene body regions with significant changes in HIF1A$^{-/-}$ cells in normoxia (Supplementary Fig. 2f, g, Supplementary Data 1).

**Fig. 1 Identification of an acute transcriptional response to hypoxia driven by HIF1A. a** Genome browser view of PRO-seq, ChIP-seq, and RNA-seq signals across the *ENO1* locus for HCT116 wild-type (WT) and HIF1A$^{-/-}$ cells exposed to normoxia (21% O$_2$, blue) or hypoxia (1% O$_2$, red). **b** Western blot analysis of HIF1A and HIF2A levels in HCT116 wild-type cells exposed to 21% or 1% O$_2$ for the indicated times. Images are representative of at least two replicates with similar results. Source data are provided as a Source Data file. **c** DESeq2 differential expression analysis of PRO-seq signal within gene body regions for HCT116 WT (blue) and HIF1A$^{-/-}$ (green) cells. Horizontal dashed lines indicate an FDR threshold of 10% for negative binomial Wald test; numbers and colored points indicate significant genes at this threshold. **d** Comparison of fold changes in PRO-seq gene body signal induced by 90 min hypoxia in HCT116 WT versus HIF1A$^{-/-}$ cells. Red/green points denote genes with significant up/down regulation; points representing all other genes are colored by density. **e** Gene set enrichment analysis (GSEA) of "Hallmark" gene sets against genes ranked by gene body fold changes induced by 90 min hypoxia in HCT116 WT cells. Red/green points denote gene sets with significant (FDR 10%) positive/negative enrichment. **f** Distributions of fold changes in PRO-seq signal at gene bodies and transcription start sites (TSS) and pausing index (PI) for upregulated genes in HCT116 WT (blue) and HIF1A$^{-/-}$ (green) cells. Horizontal spread of data points is proportional to density; boxes indicate medians and upper and lower quartiles; numbers above the HIF1A$^{-/-}$ plots indicate effect size estimates and *p*-values for two-sided Mann–Whitney U tests against the corresponding measure in WT cells. **g** Metagene showing typical PRO-seq signal profiles across upregulated genes in HCT116 WT (blue) and HIF1A$^{-/-}$ (green). Data are represented as splined linear fit lines with 95% confidence intervals in gray. **h** Upregulated genes ranked by WT/HIF1A$^{-/-}$ fold-change in normoxia, comparing the effect of HIF1A in normoxia (green/black/red points) and hypoxia (gray points). **i** Heatmap of gene body RPKM Z-scores for select example genes in HCT116 WT and HIF1A$^{-/-}$ cells in normoxia and hypoxia. See also Supplementary Figs. 1, 2 and Supplementary Data 1, 2.

These observations led us to test if basal levels of HIF1A contribute to the expression of hypoxia-inducible genes during normoxia. Indeed, we found that a subset of 107 acute hypoxia-inducible genes showed lower basal expression in HIF1A$^{-/-}$ cells (Supplementary Fig. 2h), including well-characterized HIF1A targets such as *ETS1*, *ELF3*, and *DDIT4* (Fig. 1h–i, Supplementary Data 1). Other key HIF1A targets did not show this basal HIF1A-dependence in normoxia, with some displaying lower transcription in WT cells, such as the glucose transporters *SLC2A1* and *SLC2A3* (Fig. 1h, i). These observations are consistent with low but measurable HIF1A activity in normoxia, which is further supported by analysis of CRISPR genetic screen data under normoxic conditions (see later).

**Acute transactivation associates with strong HIF1A binding.** We next investigated the relationship between acute activation and repression, and HIF1A chromatin binding as measured by ChIP-seq (Supplementary Data 3). Expectedly, HIF1A ChIP-seq peaks identified in this data set are enriched for hypoxia response element (HRE) motifs (Supplementary Fig. 3a, b) and essentially all display increased signal in hypoxia (Fig. 2a). Although ChIP-seq enrichment signal increases at TSSs for all classes of genes upon hypoxia, only those that are acutely upregulated display stronger binding around their TSS during hypoxia relative to genes with non-significant (n.s.) differences in transcription, with HIF1A peaks being most frequent within 2.5 kbp of the TSS (Fig. 2b, Supplementary Fig. 3c). Although hypoxia induces HIF1A binding near hundreds of promoter regions, peaks associated with acutely transactivated genes tend to display a stronger enrichment signal relative to either repressed or unaffected genes (Fig. 2c). Furthermore, HIF1A enrichment signal at TSS or at peaks within 50 kbp tends to be higher for genes that depend on HIF1A for their upregulation during hypoxia than for genes that display HIF1A-independent upregulation (Supplementary Fig. 3d, e). When HIF1A peaks are classified as "proximal" and "distal" according to enrichment signal and distance from TSS (Supplementary Fig. 3f), the high-confidence proximal peaks are significantly overrepresented near upregulated (~36%) but not downregulated (~10%) genes (Supplementary Fig. 3g). Earlier studies suggest that HIF1A can also operate over larger distances by acting on preformed enhancer-promoter interactions[26,27]. Our data support these findings, with distal HIF1A binding sites also being significantly overrepresented near acute upregulated (~21%) but not downregulated (~15%) genes or genes with no significant change during acute hypoxia (Supplementary Fig. 3h). Closer examination of distal HIF1A peaks revealed that those associated with upregulated genes, referred herein as "productive"

binding sites, exhibit bidirectional transcriptional activity even in normoxia, prior to upregulation of their associated genes (Fig. 2d). Bidirectional transcription at intergenic regions is a recognized hallmark of active enhancers[28–30]. Transcription at these sites increases upon exposure to hypoxia and is dampened in HIF1A$^{-/-}$ cells (Fig. 2d, e). By contrast, distal peaks not associated with upregulated genes, referred herein as "unproductive" binding sites, display much lower transcriptional activity that is mostly unaffected by hypoxia or the absence of HIF1A (Fig. 2d, e). When we examined available ENCODE data for HCT116 cells, we found that productive distal sites display increased DNaseI accessibility and increased enrichment of histone modifications associated with enhancers (e.g. H3K27ac, H3K4me1) relative to unproductive sites (Fig. 2f).

HIF1A enrichment signal for both proximal and distal peaks is positively correlated with fold changes in gene body transcription at upregulated genes, with a stronger correlation being observed for proximal peaks (Fig. 2g, h). Overall, genes associated with proximal and/or distal HIF1A peaks account for ~52% of acute transactivation events (Fig. 2i) and genes with both peak types tend to display larger increases in productive transcription (Fig. 2j). Together, these data indicate that HIF1A binding is associated with acute transactivation, and do not support a direct role for HIF1A in acute transcriptional repression, which may be explained in part by indirect repression of the MYC transcriptional program via MXI1 induction[22] (Fig. 1d, e, Supplementary Fig. 1d, f). Thus, from this point forth, we focused on the acute transactivation response.

Curiously, nearly half of the acute transactivation events exhibit dependence on HIF1A despite having no associated HIF1A peaks within 50 kbp (Fig. 2i, j, "no peak" category), suggesting either that some indirect transactivation events can occur by 90 min of hypoxia, or that these genes are regulated by HIF1A enhancers located at much greater distances. In support of the former hypothesis, IPA upstream regulator analysis predicts that some of these genes are regulated by signaling pathways, transcription factors, and chromatin regulators that are themselves directly induced by HIF1A, such as TGFB signaling (which could be explained by the strong induction of the SMAD family members SMAD3 and SMAD9), FOXO3, and KDM3A (Fig. 2k, l, Supplementary Fig. 3i). Therefore, the bulk of the acute transactivation response dependent on HIF1A can be attributed to strong HIF1A binding near direct target genes and the indirect action of early targets of HIF1A.

**CDK8 kinase activity is required for acute transactivation.** We previously identified the Mediator-associated kinase CDK8 as a transcriptional cofactor of HIF1A[31]. However, our studies

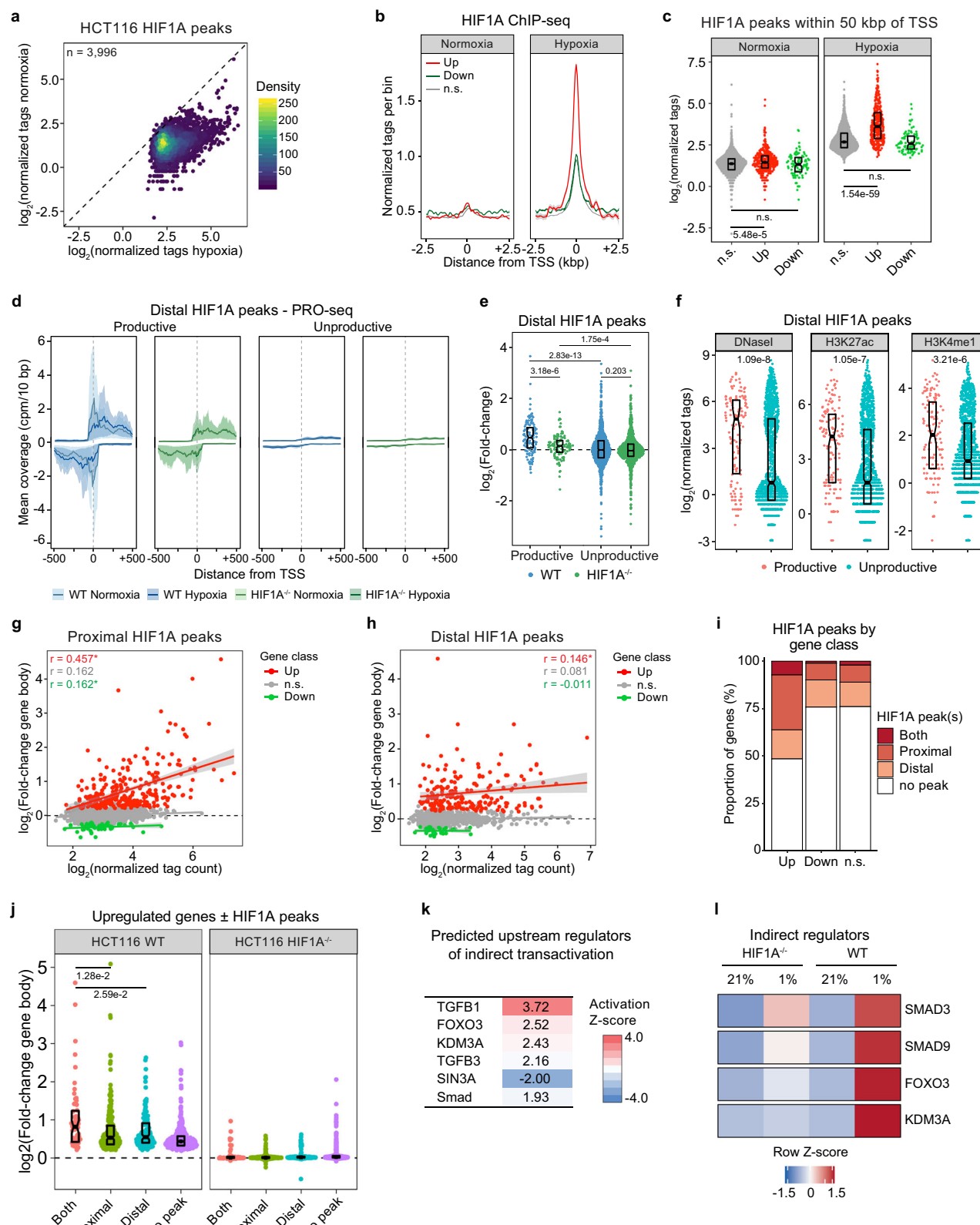

employed measurements of steady-state mRNA, which could not properly define direct versus indirect contributions of CDK8 to the transcriptional program during acute hypoxia. Furthermore, our previous studies did not test the role of CDK8 catalytic activity versus structural or scaffolding functions in HIF1A-driven transactivation. Importantly, in different settings, the

kinase activity of CKD8 has been involved in both transcriptional repression and activation[32,33], and Mediator-associated kinases have been shown to have kinase-independent effects in some transcriptional networks[33,34]. Therefore, we employed PRO-seq analysis of HCT116 cells engineered to express an "analog-sensitive" allele of CDK8 (CDK8^{as/as}) that can be specifically

**Fig. 2 Acute transactivation, not repression, is associated with HIF1A binding. a** Comparison of enrichment signal in normoxia and hypoxia for HIF1A peaks called in hypoxic HCT116 wild-type (WT) cells ($n = 3996$ peaks). Points are colored by density. **b** Meta profile showing typical HIF1A occupancy profile over transcription start sites (TSS) of upregulated (red), downregulated (green), and not significantly regulated (n.s., gray) genes during acute hypoxia. Data are represented as splined linear fit lines with 95% confidence intervals in gray. **c** Enrichment signal for HIF1A peaks within 50 kbp of TSS, separated by gene body differential expression class: not significantly regulated (n.s., gray), upregulated (red), downregulated (green). Horizontal spread of data points is proportional to density; boxes indicate medians and upper and lower quartiles; horizontal bars with numbers indicate FDR-adjusted $p$-values for two-sided Mann–Whitney U tests. **d** Meta profiles showing typical PRO-seq signal profile across distal HIF1A peak regions associated with genes upregulated (Productive) or not upregulated (Unproductive) during acute hypoxia in HCT116 WT (blue) and HIF1A$^{-/-}$ (green) cells. Data are represented as splined linear fit lines with 95% confidence intervals in lighter shades, with positive and negative values reflecting signal density on $+$ and $-$ genomic DNA strands, respectively. **e** Distributions of fold changes in PRO-seq signal ($+/-$ strands combined) within ±250 bp of distal HIF1A peaks, separated by association with genes upregulated (Productive) or not upregulated (Unproductive) during acute hypoxia in HCT116 WT (blue) and HIF1A$^{-/-}$ (green) cells. Horizontal spread of data points is proportional to density; boxes indicate medians and upper and lower quartiles; numbers indicate FDR-adjusted $p$-values for two-sided Mann–Whitney U tests. **f** Distributions of DNaseI accessibility or ChIP-seq signal enrichment for various chromatin marks within regions corresponding to distal HIF1A peaks associated with genes upregulated (Productive, pink) or not upregulated (Unproductive, teal) during acute hypoxia. Horizontal spread of data points is proportional to density; boxes indicate medians and upper and lower quartiles; numbers indicate $p$-values for two-sided Mann–Whitney U tests. **g, h** Comparison of PRO-seq (gene body) fold-change with ChIP-seq peak signal for proximal (g) and distal (h) HIF1A peaks. Gene class indicates PRO-seq gene body differential expression: not significantly regulated (n.s., gray), upregulated (red), downregulated (green). Solid lines in corresponding colors denote linear model fits to the data, with 95% confidence intervals in gray. Numbers in upper left are Pearson correlation coefficients in corresponding colors; * denotes significant correlations ($p$-value < 0.05). **i** Proportions of genes in each class associated with proximal and/or distal HIF1A peaks. **j** Distributions of fold changes in gene body activity for upregulated genes associated with different classes of HIF1A peaks: proximal and distal (Both, pink), proximal only (green), distal only (teal), or no peak (purple). Horizontal spread of data points is proportional to density; boxes indicate medians and upper and lower quartiles; numbers indicate FDR-adjusted $p$-values for two-sided Mann–Whitney U tests. **k** Heatmap of activation Z-scores for upstream regulator predictions by the Ingenuity Pathway Analysis suite for acutely upregulated genes lacking associated HIF1A binding. **l** Heatmap showing relative expression of putative indirect regulators of the acute hypoxic response. Data are represented as row-wise Z-scores calculated from RPKM values. See also Supplementary Fig. 3 and Supplementary Data 3.

inhibited by bulky ATP analogs. The generation and characterization of this cell line were previously reported[35]. Notably, the CDK8$^{as/as}$ alleles behave as hypomorphs, showing decreased kinase activity even in the absence of bulky ATP analogs, while the remaining kinase activity can be fully blocked by the ATP analog 3MB-PP1 (ref. [35]). We thus repeated our PRO-seq analysis in HCT116 WT and CDK8$^{as/as}$ cells treated with vehicle (DMSO) or 3MB-PP1 and exposed to 21% or 1% $O_2$ for 90 min (Supplementary Data 4, see Supplementary Fig. 4a, b for evaluation of biological replicates). Analysis of gene body transcription activity in DMSO- and 3MB-PP1-treated HCT116 cells revealed a widespread requirement for CDK8 kinase activity for hypoxia-driven transactivation, with a more obvious impact in the +3MB-PP1 conditions (Fig. 3a, Supplementary Fig. 4c, d). Among 1547 genes significantly transactivated upon hypoxia in this experiment, 405 were expressed at significantly lower levels upon full CDK8 inhibition with 3MB-PP1, while 99 genes showed increased expression (Fig. 3b, Supplementary Fig. 4c). Thus, among hypoxia-inducible genes, CDK8 behaves mostly as a positive regulator of transcription. This requirement for CDK8 kinase activity was also evident, albeit with milder quantitative effects, in DMSO-treated cells expressing the hypomorph CDK8$^{as/as}$ alleles (Supplementary Fig. 4c, d).

Pathway analysis reveals stronger enrichment of gene signatures associated with hypoxia, glycolysis, and extracellular matrix remodeling among hypoxia-inducible genes that require CDK8 kinase activity (Fig. 3c). Upstream regulator analysis identifies enrichment of genes regulated by HIF1A among those that required CDK8 for transactivation, whereas no clear upstream regulators are identified among "CDK8 repressed genes" (Fig. 3d). CDK8-dependent genes include prominent HIF1A targets involved in glycolysis (Fig. 3e), consistent with our previous finding that CDK8 inhibition impairs glycolysis[35]. Among hypoxia-inducible genes that require CDK8 kinase activity for transactivation, CDK8 inhibition significantly reduced productive elongation, with lesser impacts on transcription at TSSs (Fig. 3f, g, Supplementary Fig. 4e, f). Thus, upon CDK8 inhibition, pausing indices were not decreased at these genes during hypoxia,

reinforcing the notion of HIF1A and CDK8 working coordinately to stimulate RNAPII elongation (Fig. 3g).

**Interplay between acute transactivation and mRNA expression.** We next examined the relationship between the acute transactivation response and subsequent changes in steady-state mRNA levels at 24 h hypoxia (Supplementary Data 5). A majority (~59%) of acute transactivation events correspond to significant increases in steady-state polyadenylated mRNAs at the later time point (460 genes, Fig. 4a). We define this gene set henceforth as acute response genes. Expectedly, hypoxia induced many more changes at the mRNA level at the later time point that did not correspond to early transactivation events, and we refer to these genes as late response genes (1672 genes, Fig. 4a). We then asked how HIF1A binding related to these two groups. When probing only for proximal HIF1A binding, ~45% (209 out of 460) of acute response genes display nearby HIF1A binding, compared to only ~13% (217 out of 1672) of late response genes (Supplementary Fig. 5a). When extending the analysis to also include distal peaks, the percentages increase to ~58% (268 out of 460) for acute response genes and ~27% (457 out of 1672) for late response genes. Thus, genes that are hypoxia-inducible as observed by both PRO-seq and RNA-seq are more likely to be bound by HIF1A. Furthermore, HIF1A peaks associated with acute response genes tend to display stronger levels of HIF1A binding relative to peaks associated with late response genes (Fig. 4b). This exercise reveals that simply cross-referencing steady-state mRNA data with HIF1A ChIP-seq data would predict the existence of hundreds of putative direct HIF1A targets that are not immediately induced by hypoxia (i.e. 457 genes when using proximal + distal sites, Supplementary Fig. 5b). Additionally, this approach would miss dozens of acute response genes identified by PRO-seq for which nearby HIF1A is not evident (Supplementary Data 2).

Interestingly, the subset of transcripts that were detected as hypoxia-inducible only by PRO-seq (234 genes, Fig. 4a) included many antisense RNAs, miRNAs, and long non-coding RNAs (lncRNAs) (Supplementary Fig. 5c, d, Supplementary Data 6). Given the evolving annotation of lncRNAs, many of which have

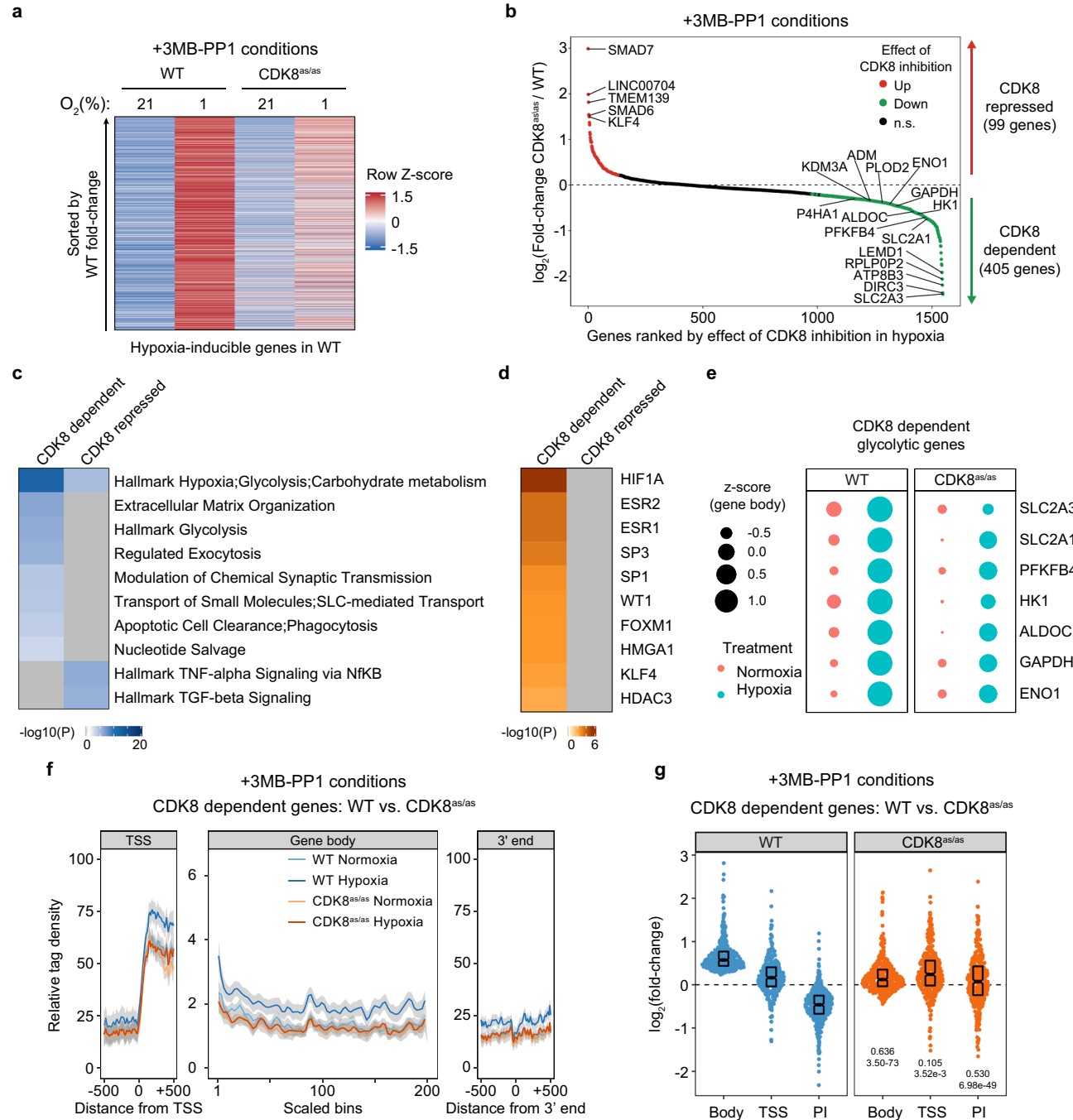

**Fig. 3 The acute transcriptional response to hypoxia requires CDK8 kinase activity. a** Heatmap showing relative gene body signal for genes with increased transcription activity after 90 min hypoxia in wild-type (WT) and CDK8$^{as/as}$ HCT116 cells treated with 3MB-PP1 during normoxia (21% $O_2$) and hypoxia (1% $O_2$). Data are represented as row-wise Z-scores calculated from RPKM values. **b** Upregulated genes ranked by CDK8$^{as/as}$/WT fold-change in hypoxia + 3MB-PP1. Genes with significant differences in gene body activity during CDK8 inhibition are highlighted in red (increased activity) and green (decreased activity). **c** Comparison of top 10 pathway/function clusters enriched among CDK8 dependent and CDK8 repressed genes, as identified by Metascape. Heatmap color represents $-\log_{10}$(p-value) from hypergeometric enrichment test. **d** Comparison of transcription regulators with enriched targets among CDK8-dependent and CDK8-repressed genes, as identified by the TRRUST module of Metascape. Heatmap color represents $-\log_{10}$(p-value) from hypergeometric enrichment test. **e** Bubble plots showing relative transcription activity in WT and CDK8$^{as/as}$ HCT116 cells exposed to normoxia (pink) or hypoxia (blue) for CDK8-dependent glycolytic genes. Circle areas correspond to gene-wise Z-scores calculated from gene body RPKM values. **f** Metagene showing typical PRO-seq signal profiles across CDK8-dependent upregulated genes in HCT116 WT (blue) and CDK8$^{as/as}$ (orange) cells treated with 3MB-PP1. Data are represented as splined linear fit lines with 95% confidence intervals in gray. **g** PRO-seq fold-change distributions for gene body, transcription start site (TSS) and pausing index (PI) of CDK8-dependent upregulated genes in HCT116 WT (blue) and CDK8$^{as/as}$ (orange) cells treated with 3MB-PP1. Horizontal spread of data points is proportional to density; boxes indicate medians and upper and lower quartiles; numbers below the CDK8$^{as/as}$ plots indicate effect size estimates and p-values for two-sided Mann–Whitney U tests against the corresponding measure in WT cells. See also Supplementary Fig. 4 and Supplementary Data 4.

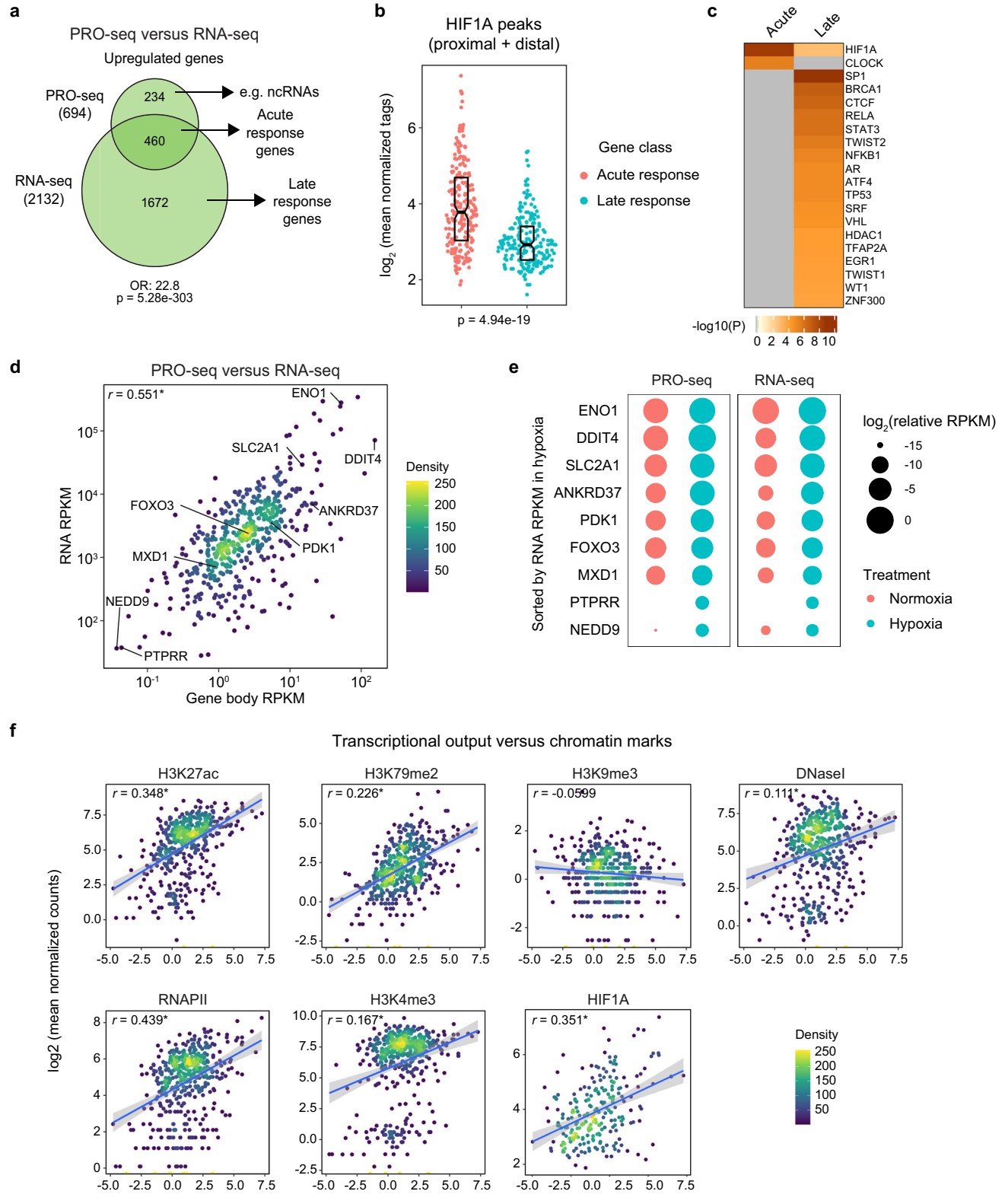

been reclassified as eRNAs upon further examination[36], we hope our PRO-seq data set will enable a more detailed characterization of these transcripts in future studies.

Upstream regulator analysis demonstrates strong enrichment of HIF1A targets among acute response genes, whereas targets of a much larger set of transcriptional regulators are enriched among the late response genes (Fig. 4c), involving many other

pathways (Supplementary Fig. 5e). From this point forth, we focused our analyses on acute response genes.

When comparing gene-body PRO-seq signals to mRNA RNA-seq signals for acute response genes, we observed that both RNA output levels and fold-change are positively correlated between the two measurements (Fig. 4d, Supplementary Fig. 5f), with many changes in transcriptional activity being amplified in terms

**Fig. 4 Acute transcriptional output shapes the steady-state transcriptional response to hypoxia. a** Qualitative overlap between genes called as upregulated in PRO-seq (90 min hypoxia) and RNA-seq data sets (24 h hypoxia). Odds ratio (OR) and p-value (p) are for two-sided Fisher's exact test. **b** Distributions of HIF1A ChIP-seq enrichment signal for HIF1A peaks associated with genes upregulated in both PRO-seq and RNA-seq data sets (Acute response, pink) or RNA-seq only (Late response, teal). Horizontal spread of data points is proportional to density; boxes indicate medians and upper and lower quartiles. Number below plot indicates p-value for two-sided Mann–Whitney U test between the two distributions. **c** Comparison of transcription regulators with enriched targets among Acute response and Late response genes, as identified by the TRRUST module of Metascape. Heatmap color represents $-\log_{10}(p\text{-value})$ from hypergeometric enrichment test. **d** Comparison of transcription activity at 90 min hypoxia (gene body RPKM) with mRNA levels after 24 h hypoxia (RNA RPKM) for Acute response genes. Points are colored by density and selected example genes are labeled. Pearson correlation coefficient is shown in upper left, * denotes significant correlation ($p < 2.2e{-}16$). **e** Bubble plots showing relative transcription activity in wild-type (WT) and CDK8$^{as/as}$ HCT116 cells exposed to normoxia (pink) or hypoxia (blue) for CDK8-dependent glycolytic genes. Circle area corresponds to $\log_2$(RPKM) values, scaled within each data set. **f** Comparison of transcription activity at 90 min hypoxia (gene body RPKM) with enrichment signal (mean normalized tag counts) at TSS regions (for ENCODE project chromatin and RNAPII data) or peak regions (for HIF1A ChIP-seq data) for Acute response genes. Points are colored by density. Blue lines denote linear model fits to the data, with 95% confidence intervals in gray. Numbers in upper left are Pearson correlation coefficients, * denotes significant correlations (10% FDR). FDR-corrected p-values are as follows: H3K27ac 4.07e−29, H3K79me2 1.29e−27, H3K9me3 5.36e−2, DNaseI 3.93e−6, RNAPII 3.70e−15, H3K4me3 1.26e−7. See also Supplementary Figs. 5, 6 and Supplementary Data 5, 6.

of steady-state changes (Supplementary Fig. 5f). However, fold changes in gene body activity have a weaker correlation with output levels (Supplementary Fig. 5g). This suggests that early transcriptional output is an important driver of steady-state mRNA levels at much later time points, with hypoxia-inducible genes displaying expression levels covering several orders of magnitude (Fig. 4d). For example, HIF1A targets such as *ENO1* and *DDIT4* produce >100-fold more nascent RNA early on and steady-state mRNA later on relative to lowly transcribed genes such as *PTPRR* and *NEDD9* (Fig. 4e). To investigate the mechanisms modulating the strength of transcriptional output, we analyzed variations in chromatin environment, RNAPII occupancy, and HIF1A binding. Transcriptional output at 90 min of hypoxia correlates positively with the pre-existing levels of histone marks associated with gene activity, such as H3K27 acetylation (H3K27ac) and H3K79 dimethylation (H3K79me2), but not so with H3K9 trimethylation (H3K9me3), a mark of repressed chromatin (Fig. 4f, Supplementary Fig. 6a). Highly transcribed acute response genes also show higher levels of DNAse I accessibility, RNAPII occupancy, and H3K4me3 (a mark of transcriptional initiation) around their promoters in normoxia (Fig. 4f, Supplementary Fig. 6a). By contrast, these chromatin features are not correlated with the fold-change in transcriptional activity (Supplementary Fig. 6b). Lastly, HIF1A peaks associated with highly transcribed genes tend to show higher enrichment signal in hypoxia (Fig. 4f). Thus, the strength of HIF1A binding correlates with both absolute transcriptional output (Fig. 4f) and fold-change in transcriptional output (Fig. 2g) among its target genes.

Altogether, these observations indicate that an open chromatin environment during normoxia is positively associated with the strength of transcriptional output (but not fold-change) during the acute response to hypoxia and with subsequent steady-state mRNA levels.

**Conservation of the acute transcriptional response to hypoxia.** To investigate the conservation of the acute transcriptional response to hypoxia, we repeated RNA-seq and HIF1A ChIP-seq analyses for three additional cancer cell types: RKO (colorectal carcinoma), A549 (lung cancer), and H460 (lung cancer) (Supplementary Data 3, 5). Although we observed high overall diversity among hypoxia-induced mRNAs detected by RNA-seq after 24 h hypoxia (Supplementary Fig. 7a), ~90% of the acute response genes identified in HCT116 cells were induced in one or more additional cell lines (Fig. 5a, b). Genes upregulated in all four cell lines (core) tend to display greater increases in both nascent RNA and steady-state mRNA relative to less-conserved (shared 3, shared 2) or HCT116-specific genes (Fig. 5c, d).

Comparison of HIF1A peaks identified in each cell type indicates that, although HIF1A binding is consistently induced by hypoxia in all four cell types (Supplementary Fig. 7b) and consistently associated with underlying HREs (Supplementary Fig. 7c), there is vast cell type-specificity in HIF1A peak calls (Supplementary Fig. 7d). However, within this massive diversity, common HIF1A peaks show stronger HIF1A binding (Fig. 5e) and those associated with core genes are mostly conserved across cell types (Fig. 5f) and display significantly higher HIF1A signals than peaks associated with non-core genes (Fig. 5g). Furthermore, HIF1A peaks are associated with greater fractions of core upregulated genes in each cell type than less-conserved or cell type-specific genes (Supplementary Fig. 7e).

Overall, these results reveal that acute response genes are more conserved across cancer cell types than late response genes, commonly associated with high-occupancy HIF1A binding sites, and thus more likely to play a role in the cellular adaptation to hypoxia across multiple cancer types.

**A tumor-suppressive role for HIF1A linked to mTOR suppression.** Having identified the acute transcriptional response to hypoxia, which is more conserved than the late response across several cancer cell lines and also more clearly associated with strong and conserved HIF1A binding events, we then set out to investigate the contribution of these acute response genes to HIF1A-dependent processes in cancer cells. First, we analyzed genome-wide CRISPR-mediated knockout data for 625 cancer cell lines from the Cancer Dependency Map project[37]. In this data set, depletion of tumor suppressor genes leads to improved cell fitness and positive gene effect scores (e.g. RB1), while depletion of oncogenes has the opposite effect (e.g. KRAS) (Fig. 6a). Gene effect scores for HIF1A are positive in most cell types with a distribution similar to that of RB1, whereas the HIF repressors HIF1AN, EGLN1, and VHL clearly promote cell fitness (Fig. 6a). Therefore, in the context of in vitro cell culture under normoxia and nutrient-rich conditions, HIF1A behaves as a strong tumor suppressor through cell-autonomous mechanisms. Furthermore, this reveals a clear cellular role for HIF1A under normoxia, likely explained by the existence of additional HIF1A-activating stimuli such as growth factor signaling and genetic alterations in cancer cells[2,3].

Next, to assess the contribution of acute response genes to HIF1A suppressive effects in vitro, we analyzed co-dependency relationships in this large data set (Supplementary Table 7). Reassuringly, the HIF1A cofactor ARNT shows a strong positive co-dependency with HIF1A, while the HIF1A repressors EGLN1-3, VHL, and HIF1AN/FIH exhibit strong negative relationships (Fig. 6b, c, Supplementary Fig. 8a), confirming the validity of this

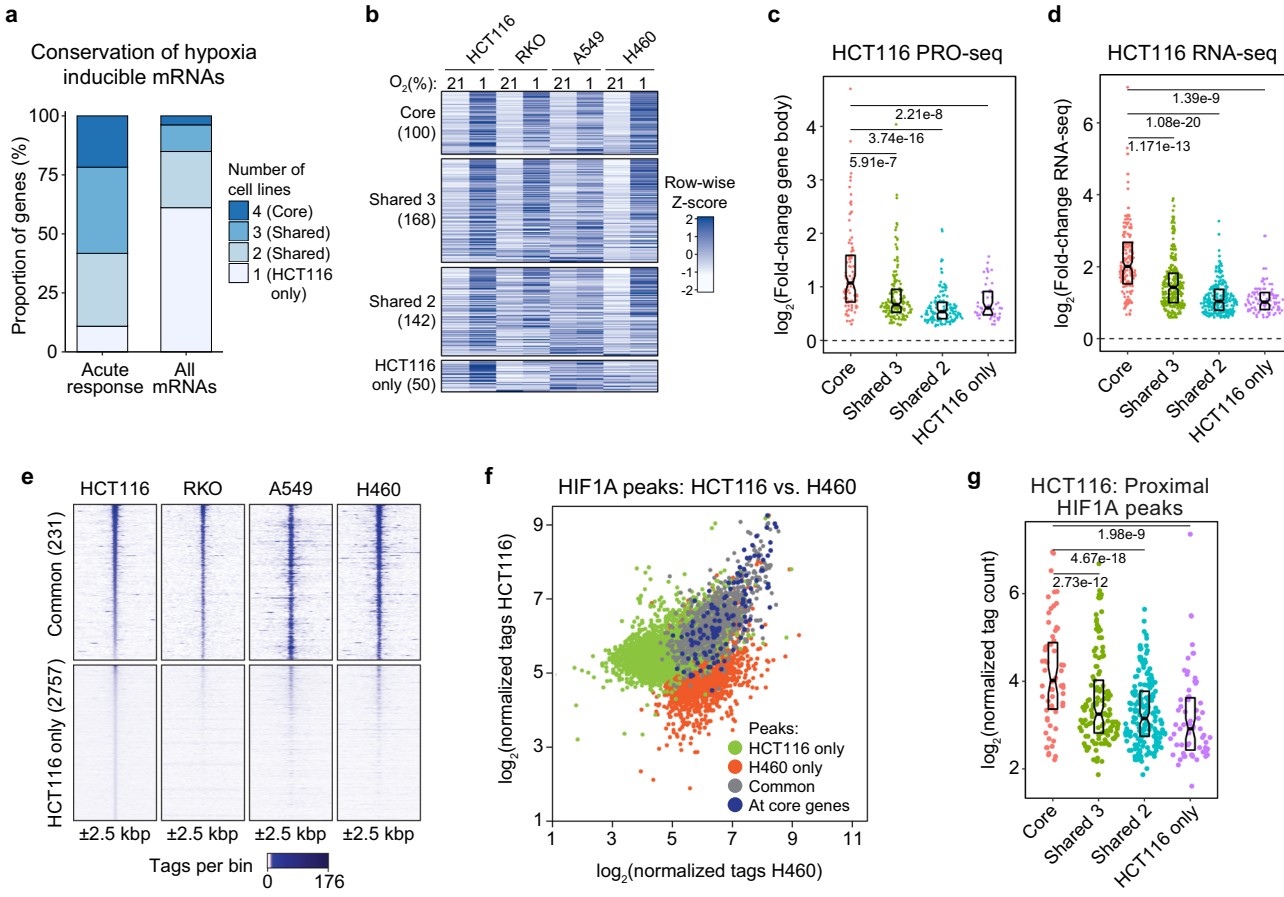

**Fig. 5 Upregulation of acute response genes by HIF1A is conserved in multiple cell types. a** Proportions of genes upregulated at both mRNA and PRO-seq levels (Acute response) or at the mRNA level (All mRNAs) in HCT116 cells with mRNA upregulation in one or more cell types. **b** Heatmap showing relative mRNA levels for acute response genes, grouped by conservation as in (**a**), across all four cell types during normoxia (21% $O_2$) or hypoxia (1% $O_2$). Data are represented as Z-scores calculated from RNA-seq RPKM values. **c, d** Fold-change distributions for gene body transcription activity (c, PRO-seq) and mRNA levels (**d**, RNA-seq) in HCT116 cells, separated by conservation group as in (**a**): core (pink), shared 3 (green), shared 2 (blue), and HCT116 only (purple). Horizontal spread of data points is proportional to density; boxes indicate medians and upper and lower quartiles; horizontal bars with numbers indicate FDR-adjusted *p*-values for two-sided Mann–Whitney U tests. **e** Heatmap of enrichment signal (±2.5 kbp, 100 bp bins) in HCT116, RKO, A549, and H460 cells exposed to hypoxia (1% $O_2$ for 24 h) for HIF1A peaks called in all four cell lines (Common) and peaks called only in HCT116 cells. Rows are sorted by HCT116 enrichment signal. **f** Comparison of enrichment signal in hypoxic HCT116 vs. H460 cells for HIF1A peaks called in one or both cell types. Colors highlight common (gray) and cell type-specific peaks (green, red) as well as peaks associated with genes with conserved upregulation in all four cell lines (core genes, blue). **g** Distributions of enrichment signal in hypoxic HCT116 cells for proximal HIF1A peaks associated with acute response genes, separated by conservation group as in (**a**): core (pink), shared 3 (green), shared 2 (blue), and HCT116 only (purple). Horizontal spread of data points is proportional to density; boxes indicate medians and upper and lower quartiles; horizontal bars with numbers indicate FDR-adjusted *p*-values for two-sided Mann–Whitney U tests. See also Supplementary Fig. 7.

approach for identifying *bona fide* functional relationships. Conversely, for VHL the strongest negative correlation is with HIF1A, while the HIF1A repressors EGLN1 and EGLN3 as well as the VHL cofactors ELOB and CUL2 exhibit positive correlations. (Fig. 6d, e, Supplementary Fig. 8b).

Notably, the strongest positive correlation for HIF1A is DDIT4 (REDD1) (Fig. 6b, c), a core acute response gene (Supplementary Fig. 8c) previously characterized as a direct HIF1A target[38]. DDIT4 was previously shown to repress mTOR signaling through the TSC1-2 complex[39], a notion that is recapitulated by our co-dependency analysis, whereby DDIT4 shows positive correlations with TSC1-2 and negative correlations with MTOR and LAMTOR1 (Fig. 6f, g, Supplementary Fig. 8d). There is also a strong positive relationship between HIF1A and FOXO3 (Fig. 6b, c, Supplementary Fig. 8a), another acute response gene that encodes a transcription factor with central roles in multiple stress responses[40], and a known repressor of mTORC1 (ref. [41]).

Altogether, these results point to cell-autonomous anti-proliferative effects for HIF1A before the onset of hypoxia, likely through suppression of mTOR signaling via induction of *DDIT4*, a target gene sensitive to basal levels of HIF1A (Fig. 1h, i).

**Acute response genes associated with cancer progression.** Having shown that acute transcriptional output and early fold changes in transcriptional activity have proportional impacts on steady-state mRNA levels at late time points of hypoxia, we then set out to investigate the prognostic value of acute response genes through analysis of ~11,000 human tumors across 27 cancer types for which gene expression are available via The Cancer Genome Atlas (TCGA) and for which curated patient survival data have been derived[42]. We first tested for association between progression-free interval (PFI) and an aggregate expression score across all acute response genes present in this data set using an iterative log-rank approach to find the optimal split between samples with high and low scores (see Methods, Supplementary Table 8).

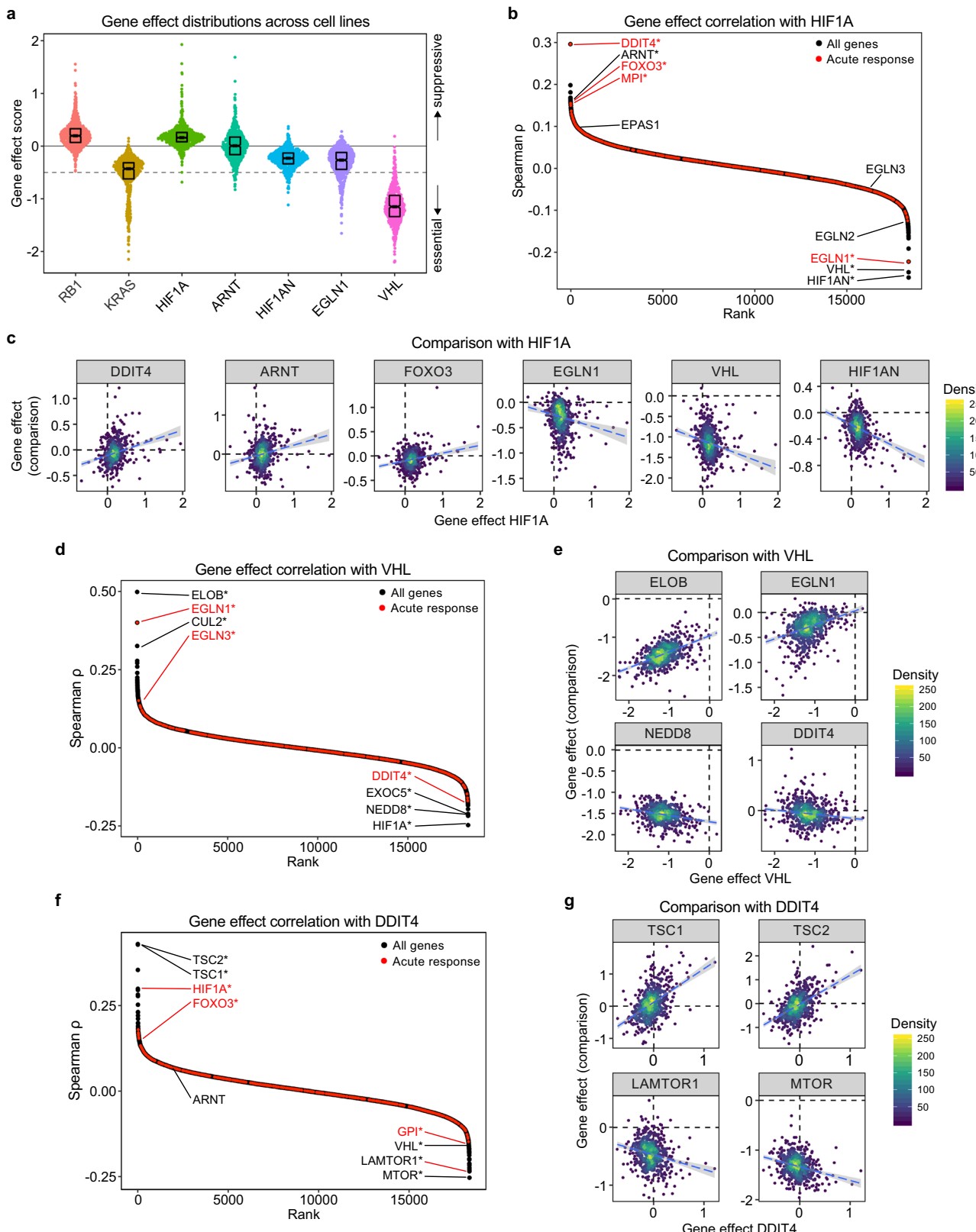

Importantly, high expression scores were predominantly associated with decreased survival, reaching statistical significance (FDR < 10%) in six cancer types: stomach adenocarcinoma (STAD), adenoid cystic carcinoma (ACC), cervical squamous cell carcinoma (CESC), low-grade gliomas (LGG), kidney renal papillary cell carcinoma (KIRP), and urothelial bladder carcinoma (BLCA)

(Fig. 7a, b, Supplementary Fig. 9a), indicating that high expression of hypoxia-inducible genes as a group is preferentially associated with worse outcome. Only one tumor type, skin cutaneous melanoma (SKCM), presented better prognosis when expressing high levels of acute response genes (Fig. 7a, Supplementary Fig. 9a). We then analyzed acute response genes individually (Supplementary

**Fig. 6 HIF1A suppresses cancer cell viability in normoxia. a** Gene effect score distributions for *HIF1A* and select known regulators, alongside the tumor suppressor *RB1* and oncogene *KRAS* for comparison, Horizontal spread of data points is proportional to density; boxes indicate medians and upper and lower quartiles; dashed line indicates threshold below which a gene is considered essential. **b** Ranked Spearman correlation coefficients of gene effect scores for each gene against *HIF1A*, with selected genes labeled, and acute response genes highlighted in red. * denotes significant correlations (FDR < 10%) — see Supplementary Data 7 for exact FDR values. **c** Comparison of gene effect scores with select genes for *HIF1A*. Points representing cell lines are colored by density and dashed lines represent linear fits to the data with 95% confidence intervals in gray. **d** Ranked Spearman correlation coefficients of gene effect scores for each gene against *VHL*, with selected genes labeled, and acute response genes highlighted in red. * denotes significant correlations (FDR < 10%) — see Supplementary Data 7 for exact FDR values. **e** Comparison of gene effect scores with select genes for *VHL*. Points representing cell lines are colored by density and dashed lines represent linear fits to the data with 95% confidence intervals in gray. **f** Ranked Spearman correlation coefficients of gene effect scores for each gene against *DDIT4*, with selected genes labeled, and acute response genes highlighted in red. * denotes significant correlations (FDR < 10%) — see Supplementary Data 7 for exact FDR values. **g** Comparison of gene effect scores with select genes for *DDIT4*. Points representing cell lines are colored by density and dashed lines represent linear fits to the data with 95% confidence intervals in gray. See also Supplementary Fig. 8 and Supplementary Data 7.

Table 8), focusing on those significantly associated with PFI in multiple cancer types (FDR < 5%). High expression of individual acute response genes was more often associated with unfavorable than favorable prognosis (161 versus 107, respectively), with some genes exhibiting divergent associations in different cancer types (83 genes) (Supplementary Fig. 9b). Of note, MXI1, the acute response gene known to repress the MYC transcriptional program, was consistently associated with favorable prognosis, as would be expected for a suppressor of MYC signaling (Supplementary Fig. 9c, d).

Importantly, when we analyzed the pathways enriched among acute response genes significantly associated with decreased PFI across increasing numbers of cancer types, we observed increased enrichment only of genes related to ECM remodeling (Fig. 7c, d). This subset of acute response genes included key enzymes involved in collagen remodeling (e.g. PLOD2, PLOD1, P4HA1, LOXL2), collagen subunits (COL1A1, COL13A1), and proteins previously involved in cell migration and invasion, such as SERPINE1 (ref. [43]), CD109 (ref. [44]), and calpastatin (CAST)[45]. Some of these genes have previously been linked to hypoxia-induced ECM remodeling, local invasion, and/or metastasis, such as *PLOD1/2*, *P4HA1*, and *LOXL2* (refs. [3,46]). When we repeated this analysis using Overall Survival (OS) as an endpoint instead of PFI, high expression of these ECM-related genes was also associated with shorter time to death (Supplementary Fig. 9e). Consistently, high expression of these genes in diverse tumor types is associated with poor prognosis (Fig. 7e, Supplementary Fig. 9d). Cox regression analysis with additional variables (see Methods) further confirmed association of these genes with adverse outcome (Fig. 7f, Supplementary Table 8). Notably, PFI and OS data may not be sufficiently mature for some cancer types in the TCGA data set, which prompts cautious interpretation of negative results in this analysis.

Overall, these results highlight the pleiotropic nature of the acute hypoxic response, with tumor-suppressive and oncogenic roles in different contexts through the action of diverse HIF1A targets, while also revealing a prominent role for hypoxia-induced ECM remodeling in cancer progression.

## Discussion

Several proteins capable of exerting $O_2$-sensitive signal transduction have been identified in humans, most prominent among them the well-characterized HIF transcription factors[47], Jmjc domain-containing lysine demethylases[8,9], and the thiol oxidase ADO[10]. The recent identification of HIF-independent $O_2$-sensing mechanisms reveals the need to dissect their relative contribution to the overall cellular and organismal responses to hypoxia, as well as their roles in cancer and other pathologies. The Jmjc domain-containing lysine demethylases KDM5A and KDM6A are inhibited during hypoxia, resulting in hypermethylation of

histone marks associated with gene activation and repression, respectively[8,9], which could potentially enforce $O_2$-sensitive epigenetic programs. The thiol oxidase ADO, a human homolog of plant cysteine oxidases, was found to regulate protein stability in an $O_2$-sensitive fashion through the N-degron pathway, including targets involved in calcium signaling, MAPK signaling, and angiogenesis[10]. These two $O_2$-sensing mechanisms may evolutionarily predate the HIFs and appear capable of rapid responses to hypoxia[48], suggesting that the earliest hypoxic signaling may be independent of the HIFs. To advance knowledge in this area, we completed a multi-omics analysis of the cellular response to hypoxia, including an analysis of nascent RNA synthesis at short time points of $O_2$ deprivation.

Our PRO-seq analysis of the acute transcriptional response to hypoxia reveals a strong dependency on HIF1A. Most acute upregulated genes harbor relatively high levels of both active/paused RNA polymerase, indicative of a permissive chromatin landscape prior to hypoxic exposure. While we cannot discount a possible role for further chromatin modification upon acute hypoxia, this suggests that neither chromatin accessibility nor RNAPII recruitment are limiting factors at most of these genes. We previously demonstrated that HIF1A induces the recruitment of elongation factors including CDK9/P-TEFb to stimulate increased expression of target genes (measured as steady-state mRNA) and that many of these genes harbor paused RNAPII in normoxia, leading us to propose a transactivation model whereby HIF1A acts predominantly by stimulating the release of pre-loaded, paused RNAPII into productive elongation[31]. We also showed that this process involves recruitment of the CDK8-Mediator complex to HIF1A targets, many of which require CDK8 for induced expression[31]. Our PRO-seq analysis of transcription in both normoxia and hypoxia reported here provides a definitive test of this model, demonstrating that hypoxia indeed stimulates the release of TSS-proximal RNAPII at HIF1A target genes, with a consequent increase in transcription across gene bodies, and that these events require the kinase activity of CDK8. Interestingly, we also found that a subset of HIF1A targets is sensitive to HIF1A status even in normoxia, a finding that became particularly relevant during our analysis of genetic screens performed in normoxia. These observations could be linked to the fact that HIF1A activity in normoxia can be increased by mere depletion of SIRT6, a histone deacetylase that functions as a negative regulator of RNAPII elongation[49,50].

Our comparison of acute transactivation events with matched HIF1A ChIP-seq data indicates that hypoxia-driven transactivation, but not repression, is a direct effect of HIF1A. Instead, repression could be explained by the indirect action of early HIF1A targets such as MXI1 (ref. [22]). Interestingly, many acute activation events are dependent on HIF1A despite the lack of any associated HIF1A binding sites. While this may partly be

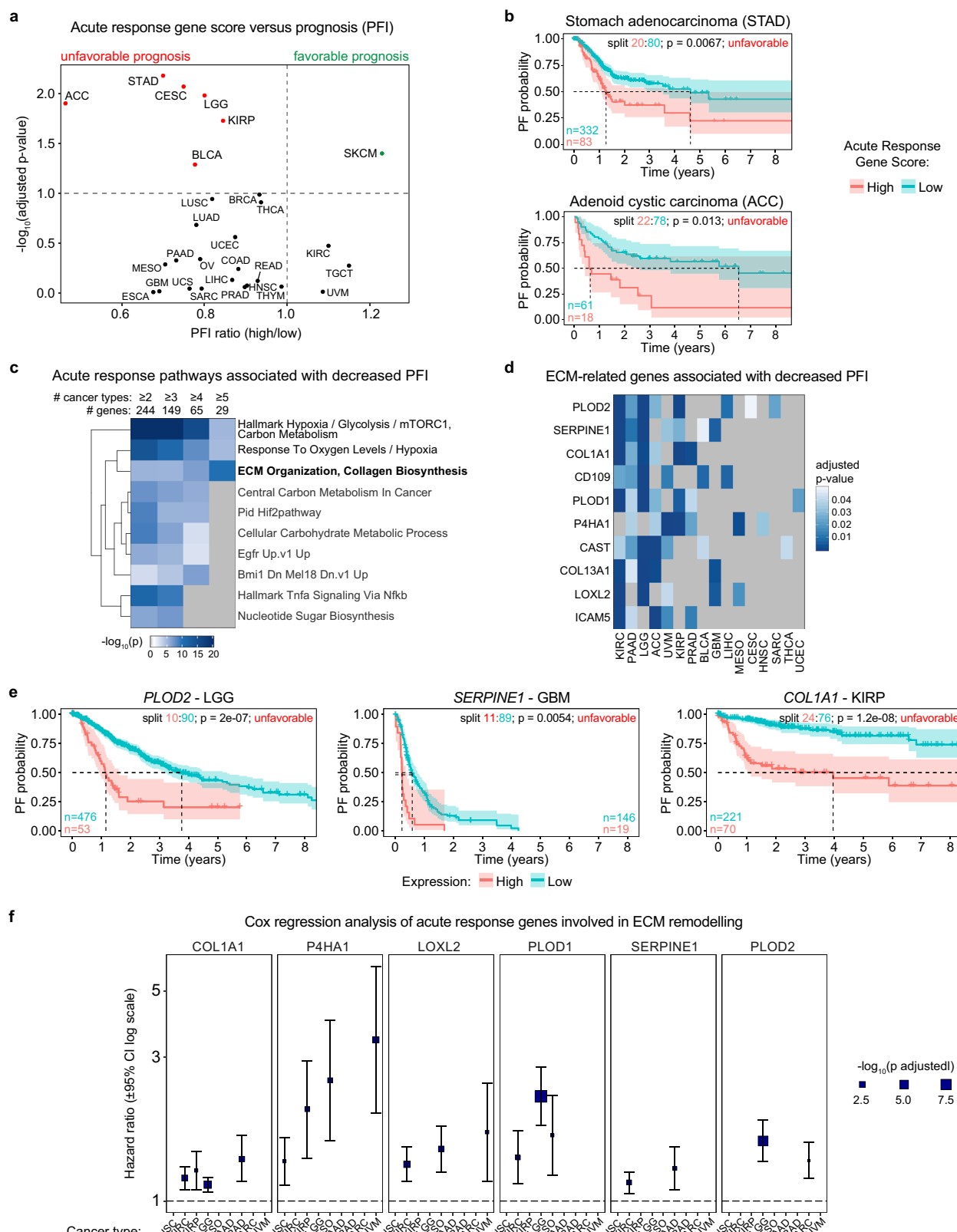

explained by false negatives due to limitations in linking HIF1A peaks to genes, our analysis of putative upstream regulators identified several factors with targets enriched in this category that are themselves direct acute HIF1A targets, such as SMAD factors and FOXO3. This suggests that secondary transcriptional events may be occurring very early during the hypoxic response

and warrants further investigation. The acute transactivation of SMAD factors, which are key mediators of TGFB signaling, is potentially relevant to the dual roles of the HIF1A network in cell-autonomous tumor suppression versus cell-extrinsic tumor promotion, as TGFB signaling is known to display similar contrasting effects at different stages of cancer progression[51]. The

**Fig. 7 Acute response genes transactivated by HIF1A involved in ECM remodeling are associated with cancer progression. a** Iterative log-rank analysis of Acute Response Gene Score vs. progression-free interval (PFI): FDR-adjusted *p*-value vs. PFI ratio (mean survival time in high group/mean survival time in low group) for the indicated cancer types; red/green denotes significant association of high score with PFI (FDR < 10%). **b** Kaplan–Meier plots for indicated cancer types, separated by Acute Response Gene Score; split indicates the proportions of samples placed into high and low scoring groups, with initial numbers at risk per arm indicated at lower left. *P*-values are from log-rank analysis with the indicated split. **c** Top 10 functional enrichment clusters from Metascape pathway analysis among individual unfavorable acute response genes, limited by significance in increasing numbers of cancer types (left to right). Heatmap color represents $-\log_{10}(p\text{-value})$ from hypergeometric enrichment test. **d** Heatmap showing significant adjusted *p*-values across cancer types for ECM-related genes associated with lower PFS. Heatmap color represents $-\log_{10}(p\text{-value})$ from hypergeometric enrichment test. **e** Kaplan–Meier plots for selected individual genes in the indicated cancer types, separated by expression level; split indicates the proportions of samples placed into high and low expression groups, with initial numbers at risk per arm indicated at lower left or right. *P*-values are from log-rank analysis with the indicated split. **f** Selected ECM-related genes with significant (FDR < 10%) association with PFI using Cox regression analysis. Blue boxes indicate estimated hazard ratio, with box size proportional to FDR-adjusted *p*-value; bars indicate 95% confidence intervals. See also Supplementary Fig. 9 and Supplementary Data 8.

---

early induction of FOXO3 may in turn be involved in cell-autonomous tumor suppression by HIF1A, as FOXO3 is considered a tumor suppressor capable of repressing mTORC1 (ref. [41]).

Our analysis of acute response genes revealed strong conservation across vastly different cancer cell types. Although hypoxia induces many cell-type-specific changes in gene expression, the core set of hypoxia-inducible genes is enriched for those that were acutely transactivated in HCT116 cells. This core set was clearly associated with strong and conserved HIF1A binding events. Thus, while HIF1A may work with different partner transcription factors to drive cell type-specific responses to hypoxia[47], it nonetheless activates a core group of genes in a variety of cellular contexts. Accordingly, we restricted our analysis of genetic screen data and human cancer samples to acute response genes, which revealed tumor suppressive versus oncogenic roles for different sets of HIF1A targets.

Our analysis of genetic screen data produced by the DepMap Project revealed a clear anti-proliferative role for HIF1A in normoxia. In addition to hypoxia, HIF1A activity is also induced by additional signals common during cancer progression, including reactive oxygen species, growth factor signaling, and mutation of oncogenes and tumor suppressors[2,3]. Therefore, we hypothesized that HIF1A knockout could modulate cell viability in the normoxic conditions used in the DepMap Project, and that functionally important acute response genes might exhibit similar essentiality profiles as *HIF1A* in these experiments. Indeed, several acute response genes are among the most strongly correlated with HIF1A in this data set. Interestingly, HIF1A behaves similarly to tumor suppressors such as RB1 in the normoxic and nutrient-rich conditions under which the screenings were carried out, with its knockout having a positive effect on cell viability. While this might initially seem counterintuitive, a critical role of HIF1A in adaptation to hypoxia involves balancing energy supply and demand, in part by promoting glycolysis, downregulating mitochondrial metabolism, and suppression of mTOR signaling[52]. In this light, activation of HIF1A by oncogenic signaling in normoxia might be expected to lead to decreased cell proliferation. In this context, the strong correlations of HIF1A with the acute response genes *DDIT4* and *FOXO3*, both negative regulators of mTOR[39,41], suggests that inhibition of proliferation via mTOR suppression may be an important function for basal levels of HIF1A. Notably, our PRO-seq analysis detected reduced basal transcription activity at the *DDIT4* locus in HIF1A$^{-/-}$ HCT116 cells. Finally, this potential tumor-suppressive role of HIF1A in normoxic cancer cells suggests caution in applying therapeutic approaches aiming to inhibit HIF1A signaling.

Although high HIF1A protein levels are an independent predictor of mortality in multiple cancer types[3], HIF1A can also exert tumor-suppressive effects[5]. Moreover, oxygen levels during cancer development are seldom static[7] and, while circulating

tumor cells may be well-oxygenated, metastases frequently occur in tissues with hypoxic niches[4], providing multiple scenarios where the acute response might influence disease progression. Our analysis of TCGA expression and survival data show that stronger expression of acute response genes as a group is more commonly associated with shorter progression-free intervals across different cancer types. For individual genes, high expression was also more commonly associated with unfavorable prognosis, although many genes also had favorable associations in certain cancer types (e.g. MXI1). By focusing our analysis on the most consistent associations across cancer types, we identified a set of acute response genes related to ECM remodeling and collagen modifications as being associated with poor prognosis in multiple cancer types. Thus, induction of these genes during the acute response could conceivably contribute to dissemination of cells from initial tumors as well as establishment at sites of metastasis.

Overall, our integrative analyses of multiple genomics data sets, together with data from over 600 cancer cell lines and 11,000 human tumor samples, indicate that HIF1A is the main driver of the acute transcriptional response to hypoxia, and that this transcriptional program encodes divergent functions linked to intrinsic suppression of cancer cell proliferation as well as cell-extrinsic disease progression via ECM remodeling.

## Methods

See Supplementary Table 1 for a list of reagents and resources, including primer sequences.

**Cell culture.** HCT116, RKO, A549, and H460 cells were cultivated in McCoy's (HCT116), DMEM (RKO, A549), and RPMI media (H460) (Gibco, Thermo Fisher Scientific) supplemented with 10% fetal bovine serum (Peak Serum) and 1% antibiotic-antimycotic mixture (Gibco). HCT116 HIF1A$^{-/-}$ cells were created by disrupting exons 3 and 4 of the *HIF1A* locus using adeno-associated virus-mediated homologous recombination, resulting in a 226-bp deletion with translation stop codons in all three reading frames[53]. All cells were plated the day before treatment and maintained in a humidified atmosphere with 5% $CO_2$ at 37 °C. Cells were maintained in normoxia (as above) or exposed to hypoxic conditions by incubation in a humidified atmosphere containing 1% $O_2$, 5% $CO_2$, and 94% $N_2$ for the indicated time.

**Western blot.** Protein samples were prepared by lysing cell pellets in RIPA buffer (150 mM NaCl, 1% v/v Igepal C630 NP-40, 0.5% w/v sodium deoxycholate, 0.1% w/v SDS, 5 mM EDTA, 50 mM Tris HCl pH 8.0, and protease/phosphatase inhibitors), followed by sonication (2.5 W, 10 s), and heat-denatured in SDS-loading buffer (50 mM Tris HCl pH 6.8, 2% w/v SDS, 10% v/v glycerol, 1% 2-mercaptoethanol, 0.01% w/v bromophenol blue) for 5 min at 95 °C. Protein concentration was measured using a BCA Protein Assay Kit (Pierce, Thermo Fisher Scientific). 20 μg of total protein per sample was resolved by SDS-PAGE, electrophoretically transferred onto 0.45 μm PVDF membrane (Thermo Fisher Scientific) and blocked with 5% w/v skim milk powder in TBST buffer (10 mM Tris pH 8.0, 150 mM NaCl, 0.1% v/v Tween 20). Proteins of interest were labeled overnight at 4 °C with primary antibodies in milk/TBST. Membranes were washed three times in milk/TBST for 10 min, incubated with HRP-conjugated secondary antibodies (BioRad) for 1 h, and again washed three times in milk/TBST. SuperSignal West Pico Plus

Chemiluminescence Substrate (Pierce) was used for detection and digital images were captured using an ImageQuant LAS 4000 (GE Healthcare Life Sciences).

*Primary antibodies used for western blotting.* HIF1A: BD Biosciences BDB610959 (1:1,500); HIF2A: Cell Signaling Technology 59973 (1:1,000); alpha-tubulin: Sigma T9026 (1:10,000).

**Fluorescent detection of hypoxia in live cells.** Hypoxia in live cells was visualized using Image-iT Red Hypoxia Reagent (Thermo Fisher Scientific) according to the manufacturer's instructions. Briefly, the dye was added directly to cultivation media of HCT116 cells at a final concentration of 2.5 µM 90 min prior to hypoxia treatment. Fluorescence emission (624/40 nm) was imaged using an Olympus IX71 microscope at the indicated times.

**PRO-seq library preparation and sequencing.** PRO-seq library preparation was performed as described previously[34], based on the protocol of Mahat et al.[54].

*Preparation of nuclei.* WT and HCT116 cells were plated at a density of $6.7 \times 10^4$ per cm$^2$ in 150 mm diameter plates. The next day, sub-confluent cells (~70%) treated either with DMSO (0.1% final concentration) or 10 µM 3MB-PP1 (MilliporeSigma/Calbiochem) were exposed to normoxia or hypoxia for 90 min. In total three plates per sample were harvested. After treatment, cells were washed three times with ice-cold PBS and overlaid with ice-cold lysis buffer (10 ml per 150 mm plate, 10 mM Tris–HCl pH 7.4, 3 mM MgCl$_2$, 2 mM CaCl$_2$, 0.5% v/v NP-40, 10% w/v glycerol, 1 mM DTT, 1 mM benzamidine, 1 mM sodium metabisulfite, 0.25 mM phenylmethylsulfonyl fluoride, and 4 U/ml SUPERase-In (Thermo Fisher Scientific)). Lysates were scraped from the plates and centrifuged at 1000 g for 15 min at 4 °C. Supernatants were removed, and pellets were resuspended in 1.5 ml of lysis buffer by pipetting 30 times. An extra 8.5 ml of lysis buffer was added to each sample and suspensions were centrifuged as previously. Supernatants were discarded and pellets were resuspended in 1 ml of lysis buffer. Following centrifugation at 1000 g for 5 min at 4 °C, pellets were resuspended in 0.5 ml of freezing buffer (50 mM Tris pH 8.3, 40% w/v glycerol, 5 mM MgCl$_2$, 0.1 mM EDTA, 4 U/ml SUPERase-In) and centrifuged once more at 2000 g for 2 min at 4 °C. Pelleted nuclei were resuspended in 100 µl of freezing buffer, counted and aliquoted at $1 \times 10^7$ nuclei/100 µl before snap-freezing in liquid nitrogen and stored at −80 °C. Two independent replicates of nuclei samples were prepared.

*Nuclear run-on and library preparation.* Nuclear run-on was performed essentially as described in ref. [54]. Briefly, two $1 \times 10^7$ nuclei aliquots per sample were incubated with 100 µl of $2 \times 1$-Biotin run-on reaction mix (10 mM Tris pH 8.0, 5 mM MgCl$_2$, 1 mM DTT, 150 mM KCl, 50 µM Biotin-11-CTP (Perkin Elmer), 50 µM rCTP, 250 µM rATP, 250 µM rGTP, 250 µM rUTP (Roche/Sigma Aldrich), 20 U SUPERase-In, 1% v/v Sarkosyl) for 5 min at 37 °C. Reactions were terminated by the addition of TRIzol LS (Thermo Fisher Scientific) and RNA was extracted according to the manufacturer's protocol. Next, RNA samples were heat-denatured for 40 s at 65 °C, placed on ice, and base-hydrolyzed by adding ice-cold NaOH (0.2 M final concentration). Hydrolysis was stopped with Tris-HCl pH 6.8 (0.5 M final concentration) and buffer was exchanged by running the samples through a P-30 column (BioRad). Next, samples were enriched for Biotin-11-CTP labeled RNA by incubating for 20 min with Streptavidin beads (Dynabeads M-280, Thermo Fisher Scientific) followed by two washes with High salt buffer (50 mM Tris-HCl pH 7.4, 2 M NaCl, 0.5% v/v Triton X-100), two washes with Binding buffer (10 mM Tris-HCl pH 7.4, 300 mM NaCl, v/v 0.1% Triton X-100), and one wash with Low salt buffer (5 mM Tris-HCl pH 7.4, v/v 0.1% Triton X-100). After RNA extraction with TRIzol (Thermo Fisher Scientific), the VRA3 3′ RNA adaptor was ligated to RNA fragments by incubating with T4 RNA ligase (New England Biolabs) at 20 °C for 4 h and samples were again enriched for biotin-labeled RNA. 5′ ends of the precipitated RNA were repaired with RNA 5′ Pyrophosphohydrolase followed by phosphorylation with T4 polynucleotide kinase (both from New England Biolabs). Following ligation of VRA5 5′ RNA adaptor, samples were enriched for biotin-labeled RNA a third time and subjected to reverse transcription reaction with the RP1 reverse transcription primer and Superscript III reverse transcriptase (Thermo Fisher Scientific). Next, the cDNA was PCR amplified (Phusion HF, Thermo Fisher Scientific) using barcoded primers (Illumina TruSeq Small RNA). Final cDNA libraries were cleaned-up with Ampure XP beads (Beckman Coulter) and size-selected on a Blue Pippin (200-500 bp, 2% agarose gel, Sage Science). Single-end 75 bp sequencing was performed on the Illumina NextSeq 500 platform (RTA version: 2.4.11, Instrument ID: NB501447) at the BioFrontiers Sequencing Facility at the University of Colorado Boulder.

**PRO-seq data analysis.** At active protein-coding genes, PRO-seq generates a high density of reads near the transcriptional start site (TSS), corresponding to transcriptionally engaged RNAPII undergoing promoter-proximal pausing, a rate-limiting step at most protein-coding genes in human cells[16,54]. Read density throughout gene bodies is typically lower and reflects productive transcription elongation that contributes to mRNA pools.

PRO-seq data yield was ~61-82 × 10$^6$ raw reads and ~41-65 × 10$^6$ final mapped reads per sample. Reads were demultiplexed and converted to fastq format using

bcl2fastq (bcl2fastq v2.20.0.422) and data from two sequencing runs merged per sample. Data quality was assessed using FASTQC (v0.11.5, https://www.bioinformatics.babraham.ac.uk/projects/fastqc/) and FastQ Screen (v0.11.0, https://www.bioinformatics.babraham.ac.uk/projects/fastq_screen/). Trimming and filtering of low-quality reads was performed using BBDUK from BBTools (v37.99)[55] and FASTQ-MCF from EAUtils (v1.05, https://expressionanalysis.github.io/ea-utils/). Alignment to the human reference genome (GRCh37/hg19) was carried out using Hisat2 (v2.1.0)[56] in unpaired, no-spliced-alignment mode with a GRCh37/hg19 index, and alignments were sorted and filtered for mapping quality (MAPQ > 10) using Samtools (v1.5)[57]. Gene-level count data for TSS (−50 to +500) and gene body (+1001 to end) regions were obtained using featureCounts from the Subread package (v1.6.2)[58] with custom annotation files for single unique TSS and gene body regions per gene. Custom annotation files with single unique TSS and gene body regions per gene were generated as follows: 1) hg19 RefSeqCurated transcript-level annotation was downloaded from the UCSC genome table browser (09-07-2018), transcripts shorter than 1500 bp and non-standard chromosomes were removed, and only transcripts with unique start/stop coordinates per gene were retained; 2) Sense and antisense counts were tabulated and each candidate TSS region was ranked by sense and antisense reads to obtain a single 'most-active' TSS per gene; 3) Finally, per gene, the TSS was combined with the shortest gene body to avoid the influence of alternative transcription termination/polyadenylation sites. "End of the gene" is defined by the cleavage/polyadenylation site corresponding to the shortest version of the annotated gene according to the RefSeq annotation. The choice of filtering genes shorter than 1500 bp enables separation of TSSs versus gene body regions of sufficient length, while also removing rRNA gene from consideration. Differential expression analysis of gene body regions was assessed using the DESeq2 package (v1.22.1)[19] with a custom R script (R v3.5.1/RStudio v1.1.453/Bioconductor v3.7) and cutoffs as described in text and figure legends. Analysis of RNAPII pausing was carried out using a custom R script (R v3.5.1/RStudio v1.1.453) with the ggplot2 package (v3.1.0) used for visualizations. Gene body and TSS counts were normalized by counts-per-million and by region length (cpm/bp) and Pausing Index (PI) calculated as the ratio of normalized reads in the TSS (cpm/bp) to normalized reads in the gene body (cpm/bp). Genes with < 0.5 cpm in all samples were excluded from analysis. Means of duplicate values were used for plots and Wilcoxon/Mann–Whitney U tests. For genome browser snapshots, aligned reads were downsampled to the lower aligned read count per replicate using Samtools, to ensure equal contributions from each replicate, followed by merging of replicates and generation of coverage tracks in the bedGraph format using HOMER (v4.9.1)[59] Genome browser snapshots were generated from bedGraph files using IGV (v2.8)[60] or a custom R script (R v3.5.1/RStudio v1.1.453/Bioconductor v3.7) and the Gviz package (v1.26.4)[61].

PRO-seq data are available under GEO accession GSE145567.

**RNA-seq library preparation and sequencing.** RKO, A549, and H460 cells were plated and treated for 24 h as described above, followed by harvesting in ice-cold PBS. Total RNA was extracted from cell pellets using TRI Reagent (Sigma-Aldrich), according to the manufacturer's instructions. RNA quality was assessed using Bioanalyzer RNA 6000 Pico chips (Agilent). Poly-A( + ) RNA enrichment and strand-specific library preparation were carried out using a TruSEQ mRNA library prep kit (Illumina). Single-end 150 bp sequencing was carried out on the Illumina HiSeq 4000 platform by the Genomics Core facility at the University of Colorado Anschutz.

**RNA-seq data analysis.** RNA-seq data and processing for HCT116 cells were published previously[62] and are available under GEO accession GSE68297.

For RNA-seq data from RKO, A549, and H460 cells, data yield was ~49-65 × 10$^6$ raw reads and ~42-57 × 10$^6$ final mapped reads per sample. Data quality was assessed using FASTQC (version 0.11.2, https://www.bioinformatics.babraham.ac.uk/projects/fastqc/) and FastQ Screen (v0.4.4) was used to check for common sequencing contaminants (https://www.bioinformatics.babraham.ac.uk/projects/fastq_screen/). Low-quality bases (Q < 10) were trimmed from the 3′ end of reads and reads <30 nt after trimming were removed using the Fastx toolkit (v0.0.13.2). Reads were aligned to a GRCh37/hg19 Human reference using TopHat2 (v2.0.13, --b2-sensitive --keep-fasta-order --no-coverage-search --max-multihits 10 --library-type fr-firststrand)[63] with the UCSC hg19 GTF annotation file provided in the iGenomes UCSC hg19 bundle (https://support.illumina.com/sequencing/sequencing_software/igenome.html). Aligned reads were then filtered to remove low-quality mapped reads (MAPQ < 10) using SAMtools (v0.1.19). Alignments were then coordinate sorted, and duplicates were marked using Picard (v1.129). Quality assessment of final mapped reads was conducted using RSeQC (v2.6)[64]. Gene-level counts were obtained using HTSeq (v0.6.1)[65] with the following options (--stranded=reverse --minaqual=10 –type=exon –idattr=gene_id --mode=intersection-nonempty) using the iGenomes UCSC hg19 GTF annotation file. Differential gene expression was evaluated using DESeq2 (version 1.6.3)[19] in R (version 3.1.0), using q < 0.1 (FDR < 10%) and fold-change > 1.5 (Up) or <1/1.5 (Down) as cutoffs for differentially expressed genes. Genome browser snapshots were generated from bedGraph files using IGV (v2.8)[60].

RNA-seq data for RKO, A549, and H460 are available under GEO accession GSE145108.

**ChIP-seq library preparation and sequencing.** Sub-confluent cultures of each cell line (HCT116, RKO, A549, and H460) were placed in normoxic or hypoxic conditions for 24 h. After treatment, media was removed and replaced with 1% formaldehyde in PBS (equilibrated by bubbling for 20 min with a mix of 1% $O_2$, 5% $CO_2$, and 94% $N_2$ for hypoxia samples). To avoid rapid degradation of HIF1A, a crosslinking time of 5 min was used for all samples and terminated by the addition of glycine to 0.125 M final concentration. Following 5 min of formaldehyde quenching, plates were placed on ice and washed three times with ice-cold PBS. Subsequently, cells were lysed in RIPA buffer and sonicated to generate DNA fragments of ~200-300 bp (Qsonica Q800R, 70% amplitude, 30 s on/30 s off cycle, 20 cycles for H460 lysates, 25 cycles for HCT116 and RKO lysates, and 30 cycles for A549 lysates). Samples were centrifuged at 20,000 g for 20 min at 4 °C and protein concentration in collected supernatants was measured with a BCA Protein Assay Kit. At this step, input samples (50 μg of total protein per cell line) were set aside. For each sample, four 1 ml aliquots each containing 1 mg of total protein were used for immunoprecipitation. Samples were pre-cleared with 15 μl of RIPA-washed Dynabeads Protein G (Invitrogen, Thermo Fisher Scientific) by rocking for 1 h at 4 °C. The supernatant was then collected and incubated with 30 μl of Dynabeads and 5 μl of anti-HIF1A antibody (Novus Biologicals, NB100-134) incubated overnight on a rocker at 4 °C. Next, beads were washed twice with RIPA buffer, twice with IP wash buffer (500 mM LiCl, 100 mM Tris pH 8.5, 1% v/v NP-40, 1% w/v sodium deoxycholate), and twice with RIPA (2 min on rocker at 4 °C for each washing step). Immunocomplexes were eluted by resuspending beads in 100 μl TE buffer and 200 μl of elution buffer (70 mM Tris pH 8, 1 mM EDTA and 1.5% w/v SDS) and incubating for 10 min at 65 °C. Both eluted immunocomplexes and input samples were combined with NaCl solution to a final concentration of 200 mM and incubated overnight at 65 °C to reverse formaldehyde crosslinks, followed by treatment with 20 μg proteinase K. DNA fragments were recovered using phenol/chloroform extraction followed by ethanol precipitation and re-dissolved in TE buffer. DNA fragments were size-selected (80-600 bp) by agarose gel electrophoresis (2% gel, BluePippin) and used for barcoded library preparation with the NEBNext Ultra II DNA kit, according to the manufacturer's instructions (New England Biolabs). Final libraries were size-selected (200-600 bp, BluePippin) and analyzed on Bioanalyzer High Sensitivity DNA chips (Agilent) to confirm 200 to 400 bp fragment size range. Single-end 150 bp sequencing of pooled barcoded libraries was carried out on the Illumina HiSeq 4000 platform by the Genomics Core facility at the University of Colorado Anschutz.

**ChIP-seq data analysis.** ChIP-seq data yield was ~52-93 × $10^6$ raw reads and ~19-46 × $10^6$ final mapped reads per sample. Data quality was assessed using FASTQC (v0.11.5) and FastQ Screen (v0.11.0). Trimming and filtering of low-quality reads were performed using FASTQ-MCF from EAUtils (v1.05). Alignment to the human reference genome (GRCh37/hg19) was carried out using Bowtie2 (v2.2.9)[66] in --sensitive --end-to-end mode with a GRCh37/hg19 index, and alignments were sorted and filtered for mapping quality (MAPQ > 10) using Samtools (v1.5)[57]. Alignments were then coordinate sorted, and duplicates were marked using Picard (v2.9.4). Quality assessment of final mapped reads was conducted using RSeQC (v2.6.4)[64].

The Homer suite (version 4.3)[59] was used for the identification of peak regions, annotation, and motif enrichment analysis. HIF1A peaks were called using the findPeaks module in factor mode with input genomic DNA controls, a local fold-change threshold of 3 (-L 3) and a false-discovery rate threshold of 0.1% (-fdr 0.001). The total number of normalized sequencing tags associated with peaks common to all four lines were used to control for cell line-specific IP efficiencies (tags in common peaks) by adjusting the global fold-change over control threshold for the called peaks accordingly (-F 7 (HCT116), -F 6 (RKO), -F 13 (A549), -F 14 (H460)). Peak enrichment signals were obtained using the annotatePeaks.pl module (-size given). Enrichment of both known and de novo identified sequence motifs was analyzed with the findMotifsGenome.pl module. Peak to TSS distances were calculated with the annotatePeaks.pl module and the gUtils (v0.2.0) and GenomicRanges (v1.36.0) R packages. For direct comparison of peak signals across cell types, MAnorm (v1.1.4)[67] was used to normalize and quantify read densities at all peak loci (default settings). Third party ChIP-seq data (ENCODE) were downloaded raw and processed as described above.

To associate proximal HIF1A peaks with putative target genes while minimizing false positives and negatives, we used the approach of Ouyang et al.[68], scoring peaks according to their enrichment signal and their distance from TSS regions. Peak association factors (AF) were calculated according to the following formula: $AF = pnrd*e^{dc/d0}$ where pnrd is peak normalized read density, dc is peak center distance from the TSS and d0 is a constant (500 bp). HIF1A peak-to-gene relationships with AF > 0.1 were assigned as high-confidence promoter/TSS-proximal peaks. Lower-confidence peaks ≥5 kb ≤50 from the nearest promoter/TSS were assigned as distal, and peaks >50 kbp as intergenic (remaining peaks as other). Genome browser snapshots were generated from bedGraph files using IGV (v2.8)[60].

HIF1A ChIP-seq data for HCT116, RKO, A549, and H460 are available under GEO accession GSE145157.

**GSEA and IPA analyses.** Gene set enrichment analysis (GSEA)[69] was carried out using the GSEAPreranked module on the GenePattern server (https://cloud.genepattern.org), using log2-transformed PRO-seq gene body fold changes as the

ranking metric. Putative regulators of differentially expressed genes were predicted using the Upstream Regulator Analysis module within the Ingenuity Pathway Analysis (IPA) suite (http://www.ingenuity.com). Functional enrichment analysis was carried out for the specified gene lists using Metascape (https://metascape.org).

**DepMap data analysis.** Corrected CERES gene effect scores from DepMap public release 19Q3 were downloaded from https://depmap.org/portal/download/ (Oct 10 2019)[37,70]. Pairwise Spearman correlation scores and p-values were calculated for gene effect scores of all genes against HIF1A followed by Benjamini-Hochberg correction to control for false-discovery rate, using a custom R script (R v3.6.1/RStudio v1.1.453)[71]. Correlations for each gene with HIF1A were ranked and visualized using the tidyverse and ggplot2 packages (v1.2.1 and v3.1.1).

**TCGA data analysis.** Curated, standardized clinical outcome data for TCGA patients were obtained as Supplementary Table 1 from Liu et al.[42]. Normalized RSEM RNA-seq expression data for TCGA samples were downloaded (Oct 11, 2019) from the Broad GDAC (https://gdac.broadinstitute.org/) using the firehose_get tool. Analyses were carried out using custom R scripts (R v3.6.1 / RStudio v1.1.453). For each cancer type, genes not detected in at least 50% of samples were removed.

*Acute Hypoxia Signature scoring.* Z-scores were first calculated from RSEM expression values for each gene within each cancer type. Acute hypoxia scores were calculated as the sum of Z-scores of acute upregulated hypoxia genes (ProRna Up) in each sample.

*Iterative Kaplan–Meier log-rank testing.* To find the optimal stratification of tumor samples into high and low groups, we adopted an iterative approach similar to that described in[72], using either Acute Hypoxia Score or RSEM expression values. For each high vs. low stratification starting from the $10^{th}$ vs. $90^{th}$ percentiles and proceeding in one percentile steps, log-rank tests for differences in progression-free survival (PFI) were carried out using the survminer (v0.4.6), survival (v2.44-1.1), and purrr (0.3.3) packages in R. Only tests with unique sample partitions and at least 10 events in either high or low groups were considered. Benjamini-Hochberg correction was applied to control for false-discovery rate and the partition with the lowest p-value was retained. To visualize Kaplan–Meier survival curves, plots were generated using the survminer (v0.4.6) package.

*Cox regression analysis.* To estimate the prognostic value, or change in hazard, associated with increasing expression of each gene as a continuous value and to permit adjustment for additional variables, univariate and multivariate Cox regression analysis was carried out for each detected gene within each cancer type, using the finalfit (0.9.5), furrr (0.1.0), and purrr (0.3.3) packages in R, with progression-free survival as the dependent variable, RSEM expression value as the explanatory variable of interest. For multivariate analysis, age, sex, and stage were included as additional explanatory when available. Only model fits satisfying the proportional hazards assumption were considered further. Hazard ratio plots were generated using a modified version of the hr_plot function from the finalfit package.

**Quantification and statistical analysis.** Statistical methods and analysis details are described in figure legends and in the Results and Methods sections of the manuscript.

**Reporting summary.** Further information on research design is available in the Nature Research Reporting Summary linked to this article.

## Data availability
The data that support this work are available from the corresponding authors upon reasonable request. The genomic data sets generated during this study are available at Gene Expression Omnibus database under accessions GSE145567, GSE145108, and GSE145157. Previously generated RNA-seq data are available under accession GSE68297. ENCODE data are available at https://www.encodeproject.org (accessions ENCSR000EUT, ENCSR494CCN, ENCSR000FCP, ENCSR000DTQ, ENCSR000ENM, ENCSR000EUU). Source data are provided with this paper.

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

## Acknowledgements

The results shown here are in part based upon data generated by the TCGA Research Network: https://www.cancer.gov/tcga. This work was supported primarily by NIH grant R01GM120109 and by NSF grant MCB-1817582. Additional support was provided by the Cancer league of Colorado, the Golfers Against Cancer, the Wings of Hope Pancreatic Research Foundation, and NIH grants R01CA117907 and P30CA046934.

## Author contributions

M.D.G. and J.M.E. conceived the experiments and wrote the manuscript. Z.A., H.B., and M.D.G. performed experiments. Z.A. and M.D.G. analyzed the data.

## Competing interests

The authors declare no competing interests.
