## [Peer Review File · Nature Communications]

Reviewer #1 (Remarks to the Author):

The manuscript by Andrysiak et al profiles an acute response of cancer cells to hypoxia by employing global run-on in conjunction with the use of mutant cells lacking the cognate transcription factor HIF gene. The manuscript has two parts. The first attempts to dissect the response into Hif-dependent versus Hif-independent parts, presumably arising from secondary effects. It is well argued that Hif binding at or near promoters is associated with transcription activation, not repression. A reasonable argument is also made about conservation of Hif-dependent responses among different cell types indicating a common mechanism or mechanisms of this response. The second part uses database mining showing association of HIF with either tumor suppression or oncogenicity, depending on context. Each part of the manuscript by itself has interesting observations, but as written it is not clear to me how these two parts are linked. Attempting to connect worse cancer outcomes in TCGA database with acute response targets in pro-seq experiments in vitro is potentially intriguing, but in my opinion, as currently written, does not appear to go beyond descriptive and is not sufficient to solidly link the two parts.

↪

Overall issues.

The title refers to two modules of HIF1A network, but these are not described other than in passing. Are these related to acute versus secondary response gene cohorts based on pro-seq or as oncogenic versus tumor-suppressive genes defined by database scouting? Is knowing acute response genes potentially diagnostic for cancer outcomes or not? If acute response genes do not define outcomes one way or the other, then part 1 of the manuscript does not add much toward the second part and hence my harping about the manuscript seeming to be in two pieces. This issue appears crucial to me for the manuscript's overall impact. It would be great if it can be addressed by an argument and I apologize in advance if I may have overlooked this in the existing text.

Is there a reason to believe that rapid and transient activation of acute response genes under a hypoxia pulse involves the same mechanisms and, furthermore, illuminates the same network modules as stable steady-state overexpression of these targets observed in patient samples? There may be a generalization here about mechanistically connecting acute responses and long-term expression changes that is implied, but not tested or at least discussed.

As presented, I am not fully clear about how CDK8 fits in this story. CDK8 is supposed to be globally involved in transcription as part of Mediator: does connection of CDK8 with HIF targets reflect a special relationship between CDK8 with HIF or CDK8 requirement/ involvement extends to activation in general including any other acute response? This issue might be addressed by discussion.

Is this unusual that some genes appear inhibited by CDK8? Is this a direct acute effect (and if so, through what mechanisms) or a consequence of incomplete inhibition wherein the remaining CDK8 is delivered to certain genes? This is related to the statement about genes requiring CDK8 for transactivation. Also, on the same page, what is meant by "significant impact on productive transcription and pausing indexes" needs to be spelled out: increase, decrease, etc?

Specific questions.

Pro-seqs are not possible to interpret as it is unclear what signal looks like between individual biological replicates. As a method highly prone to variability in reported pausing indexes due to different enzymes involved in generation of clonable promoter (decapping) versus downstream (PNK) RNAs, showing individual replicates +/- hypoxia is essential. This is especially due to coverage (<<100M reads) being modest by pro-seq standards. This also applies to supplemental table 710004 where base values should be shown for each replicate.

Would it be possible to show how pausing changes on repressed genes using a metaplot and show an example of a downregulated gene as a browser shot?

Given an apparent similarity in repression between wild type and Hif^{-/-} cell types, a graph similar to 1g, but with repressed genes should be shown as well.

Figure 1a (browser shot) needs a negative control locus that does not change with hypoxia. A repressed gene example should also be shown either in main or supplement.

Figure 1d. Is it surprising to see that apart from a handful of highly activated genes, there is overall a pretty good positive correlation between pro-seq gene body signal changes in wild type versus ^{-/-} cells? Does this mean that Hif1a controls only a small part of hypoxia response and the rest of the response occurs without Hif?

Figure 1i is confusing and should be shown as actual base values before and after hypoxia. This will allow one to better judge relative levels of basal expression of these genes.

CDK8 section. Having pro-seq with an inhibitor-sensitive mutant is appreciated on its own. Is there a difference in basal transcription and/or pausing indexes between wild type and mutant cells without the inhibitor?

Line 232. What is "highly" referring to when talking about commonality of acute response genes across cell lines?

Discussion. Is the first paragraph of the discussion necessary? Parts of it may belong to introduction. The discussion could be shortened as well.

Figure S2c. Could authors speculate as to why some downregulated genes have Hif1A peaks at promoters (if I am understanding this graph correctly)?

Figure S2f. How were the attributions between distant peaks and genes done for distal HIF1 peaks?

Reviewer #2 (Remarks to the Author):

In this manuscript, Andrysiak et al. have provided a deeper understanding of HIF1A mediated changes in transcription during hypoxia, and have also identified contextual roles of HIF1A in regulating cancer. To identify acute response of hypoxic stress they have measured nascent RNA transcription by PRO-seq and have identified hundreds of genes to be up- and down-regulated in a HIF1A dependent manner. They have demonstrated that activated genes have increased HIF1A binding relative to the repressed ones. Such activation by HIF1A is mediated by the release of paused Pol II and requires CDK8 kinase activity. To distinguish between acute vs late response genes they have compared the early time-point PRO-seq data with steady-state RNA-seq after 24 h of hypoxic stress, showing that the transcription of acute response genes is significantly correlated with HIF1A binding and active chromatin marks. The authors find that most of the acute response genes in HCT116 are activated across other cancer cell types and the core genes show higher HIF1A binding. They further analyzed the genome-wide knock-out data in hundreds of cancer lines, and identified a tumor suppressive role of HIF1A that could be orchestrated by DDIT4 mediated repression of the mTOR signaling process in normoxic conditions. Interestingly, the authors also identified that high-expression of the acute response genes were associated with decreased survival in multiple cancer types, suggesting that HIF1A has contrasting roles as it could act as a tumor suppressor or as a mediator of cancer progression. In summary, the authors have done an excellent job in dissecting an important biological phenomenon, and have well-utilized a high-resolution assay to probe nascent transcription in combination with ChIP-seq and other available datasets that have provided deeper insights into HIF1A dependent transcription and its roles in hypoxia and cancer. The manuscript is very clearly written and

experiments are properly executed. I believe it will be a significant contribution to the field and serve as a resource for future studies. Here are few comments that could improve the manuscript further.

1. The authors have performed PRO-seq after 90 min of hypoxic stress to identify early responsive genes but have looked at HIF1A binding after 24 h of stress. As the levels of HIF1A look similar at 90 min vs 24 h, I wonder why the authors did not do the ChIP-seq after 90 min as well. Minor point: There is a typo in Fig. 1a (Normoxia should be 21%).

2. Minor point: In line 83 the authors say that Pol II pauses at ~50-100 nt. Actually, this estimate is around 20-60 bp from the TSS (Tome et al. Nat Genet. 2018).

3. For Fig.1c it might be helpful for the reader to see some kind of a significance cut-off that determines if the genes that have altered PRO-seq signal in the HIF1A deletion line are really significant or not. Most of the changed genes in the mutant are lowly significant relative to the WT, so are they really changed? Looks like without HIF1A there is no significant alteration of transcription. It would be also useful to quantitatively distinguish the HIF1A-dependent and the -independent genes that are altered by hypoxia and check the HIF1A binding characteristics (as done in Fig.2) of the HIF1A-dependent genes. Minor point: It will be nice to see the replicate correlation plots for the PRO-seq data in the supplement.

4. It will be useful to see the global gene-body and the TSS correlations of the WT and the mutant lines to validate that global transcription is not altered upon HIF1A deletion. Also, how does the metagene plot look like for unregulated genes in WT and mutant lines in the two conditions?

5. Interestingly, the authors observe a modest increase of paused Pol II in the WT cells upon hypoxia at the upregulated genes (Fig.1g). Does this suggest that Pol II recruitment increases at the upregulated genes in a HIF1A dependent manner? Does the pause signal also go up in the downregulated genes?

6. Did the authors analyze transcriptional changes in the paused TSSs at intergenic regions to explore the idea that HIF1A could be necessary for enhancer transcription? In this context, the authors mention in the introduction that the acute response of core genes are associated with strong HIF1A enhancers. However, I did not see any distal transcriptional regulatory element/enhancer specific analysis. Are they referring to distal HIF1A binding sites as enhancers? Do these sites show divergent transcription upon hypoxia and other enhancer specific histone marks? Minor point: The authors show non-coding RNA transcription in Fig. S3 but refer to it as S4 in the text. Please check.

7. The authors have defined gene body as +1001 to end of the gene. What is the end of the gene? How far is it from the cleavage and polyadenylation site?

8. For the CDK8 analog sensitive studies I see that the authors used a DMSO control for the CDK8 as/as cell, but did not show a metagene plot for that control with respect to the WT/treated lines in the normoxic and hypoxic conditions. It will be useful to see this comparison. In line 671 did the authors mean WT, mutant and CDK8 as/as HCT116 cells? Please check. It looks like the authors did a PRO-seq with WT and mutant lines in the norm and hypoxic conditions, and then did another PRO-seq with the WT and the CDK8 as/as line in presence of DMSO or 3MB-PP1 in the above conditions. This design is not clear in the methods section (line 671).

9. The authors claim that hypoxia induces release of TSS-proximal Pol II mediated by CDK8. However, from Fig. 3f, it appears that the TSS Pol II levels are lower in the CDK8as/as treated cells relative to WT in the hypoxic condition, suggesting that CDK8 is essential for hypoxia driven Pol II recruitment/initiation as well. Please explain.

Reviewer #3 (Remarks to the Author):

In this study, the authors examine HIF1A-mediated gene transcription in cancer cell lines. Using Precision Run-On with sequencing (PRO-seq), which method only detects transcriptionally-engaged RNA polymerases, HCT116 cells were profiled for HIF1A-mediated transcription at 90 minutes of hypoxia conditions. The Pro-seq results were compared with conventional RNA seq data from multiple cell lines under hypoxia conditions for 24 hrs, where many of these 24hr changes would involve indirect effects as compared to the acute and direct changes as measured in the Pro-seq data. Chip-seq data of HCT116 and RKO defined bound targets of HIF1A. HIF1A ChIP-seq data indicates that hypoxia-driven transactivation, but not repression, is a direct effect of HIF1A. Analysis of publicly available genetic screen data produced by the DepMap Project revealed HIF1A and genes in a HIF1A network to be essential in normoxic cells. Analysis of TCGA expression and survival data showed that higher expression of acute response genes as a group is more commonly associated with shorter survival across different cancer types.

Specific comments:

- 1) It seems that the unique advantages of the PRO-seq platform (base-pair resolution and strand-specific information, rare and common nascent RNAs, unstable nascent RNAs transcribed from enhancer regions) are not really being utilized in this study. Basically, the data are collapsed into the known genes, where there have been previous and numerous expression profiling studies of hypoxia and of defining HIF1A-specific targets. The PRO-seq method is unique from the approach of other studies, but are the key advantages specific to PRO-seq able to yield new insights into HIF1A?
- 2) The PRO-seq results at 90 minutes of hypoxia are compared with RNA-seq at 24 hr. The 24hr profiles are going to have more genes differentially expressed, due to the indirect effects and longer time frame. Are the Pro-seq data finding acute and direct targets that would not have been uncovered by conventional RNA-seq? For example, if conventional RNA-seq at 90 minutes is carried out, would additional indirect effects be found at that time point? What if the RNA-seq results are overlapped with the Chip-seq results; would this filter out the indirect effects in a similar manner to using PRO-seq? Here, we would not be as interested in the total RNA levels as we would with the differential changes in response to hypoxia.
- 3) Page 7 notes that "the acute transcriptional response identified by PRO-seq also includes many genes that have not previously been linked to hypoxic signaling". Are we sure about this point? There have been many previous profiling studies of hypoxia, a number of which (Winter, Buffa, Ragnum) were recently examined in the PCAWG hypoxia paper (<https://www.nature.com/articles/s41467-019-14052-x>). It may be that many genes in the current study have not been highlighted or focused on in previous studies, but they may be found in the previous expression profiling results. If one takes the PRO-seq hypoxia genes and compares them side-by-side with results from a number of different datasets available on GEO (in a manner similar to the Figure 5b representation), does a signature truly unique to the PRO-seq data emerge?
- 4) There have been many previous studies of HIF1A binding by Chip-seq (e.g., lots of data available on GEO), though perhaps not in the HCT116 cell line. Do the observed cell type specific differences in HIF1A binding represent a novel finding or has this been previously observed?
- 5) Do the CDK8-related findings of the present study yield new insight into CDK8 and HIF1A, beyond previous studies such as the cited Galbraith, M. D. et al. study in Cell?
- 6) Using histone data from ENCODE, the authors find that HIF1A drives its core program via high-occupancy enhancers within open chromatin. Is this something that would be unique to HIF1A as compared to other transcription factors? Or would this represent new insight from previous studies

such as the cited Platt, et al. EMBO Rep paper?

7) The use of the hypergeometric test (Figure S2, Figure 4a) is most assuredly incorrect. The hypergeometric test evaluates the chance of getting the EXACT number of overlapping genes found. The correct test would be one-sided Fisher's exact or chi-squared, which evaluates the chance of getting the observed number of overlapping genes OR MORE.

8) Figure 4d compares Pro-Seq RPKM with conventional RNA-seq RPKM. It would seem that comparing differential levels (compared to normoxia) would be more appropriate to see if the HIF1A-associated changes are sustained.

9) Similar to the above point, for Figure 4f, would using differential expression (vs normoxia) instead of gene body RPKM be more relevant?

10) The DepMap results would seem to stand on their own but be somewhat disconnected from the other analyses in the paper. Would there be any enrichment or global association between the HIF1A acute targets and the DepMap-related HIF1A network?

11) For the TCGA survival analysis, the endpoint is described in the text and figures as "Progression-Free Survival". This would seem entirely incorrect. It seems more likely that Overall Survival was the endpoint. There is a big difference between the two endpoints. Most cancer types in TCGA don't have progression free survival data available.

12) For all KM plots, the numbers of patients in each arm needs to be indicated. Also, why were different splits used for different datasets?

13) A major limitation of using TCGA data for the survival analysis is that for most cancer types the survival data are not mature. For a few cancer types like kidney one can identify robust survival correlates, but for other cancer types such as lung or breast the patients have not been followed up for long enough. This would mean that many of the negative results could be the result of insufficient data rather than a true lack of association. It would be useful to also check additional public datasets (e.g. METABRIC) for which the survival data are adequate.

14) For the figure legends, whenever a p-value is represented in the figures, the specific test used to derive the p-value should be noted.

Point by point response to Reviewer's comments.

Reviewer #1 (Remarks to the Author):

The manuscript by Andrysiak et al profiles an acute response of cancer cells to hypoxia by employing global run-on in conjunction with the use of mutant cells lacking the cognate transcription factor HIF gene. The manuscript has two parts. The first attempts to dissect the response into Hif-dependent versus Hif-independent parts, presumably arising from secondary effects. It is well argued that Hif binding at or near promoters is associated with transcription activation, not repression. A reasonable argument is also made about conservation of Hif-dependent responses among different cell types indicating a common mechanism or mechanisms of this response. The second part uses database mining showing association of HIF with either tumor suppression or oncogenicity, depending on context. Each part of the manuscript by itself has interesting observations, but as written it is not clear to me how these two parts are linked. Attempting to connect worse cancer outcomes in TCGA database with acute response targets in pro-seq experiments in vitro is potentially intriguing, but in my opinion, as currently written, does not appear to go beyond descriptive and is not sufficient to solidly link the two parts.

Response: We thank the Reviewer for this constructive comment, which challenged us to better integrate the diverse findings in the paper into a more cohesive story. This body of work aims to contribute to two major pending questions in the field:

1) What are the major mediators of the direct acute transcriptional response to hypoxia? As explained in the manuscript, this question arises from the recent discovery of other oxygen sensors capable of modulating gene expression upon low oxygen. This is highly relevant in light of two recent publications in *Science* suggesting that the acute response to hypoxia involves inhibition of O₂-dependent histone/lysine demethylases and is independent of HIF1A^{1,2}. Therefore, the first part of our manuscript addresses this question with an innovative approach by measuring *direct* transcriptional events at short time points of hypoxia. The answer from our experimental approach is unequivocal: **HIF1A is a major mediator of the primary direct acute transcriptional response.**

2) What is the role of the hypoxia-inducible transcriptional program in general, and HIFs in particular, in cancer biology? This question arises from a puzzling body of literature showing both tumor suppressive and oncogenic roles for HIFs³⁻⁸. Our manuscript makes a significant contribution in this regard by analyzing the acute response genes identified in the first part of the manuscript in terms of their impacts on cell viability *in vitro* as well as expression analysis from human tumor samples. These analyses revealed context-dependent tumor suppressive and oncogenic roles for acute response genes, with a clear tumor suppressive role for HIF under normoxia linked to its target DDIT4 and mTOR suppression, and a clear oncogenic role for acute response genes involved in collagen remodeling in the context of human tumors.

Therefore, the two main efforts in the paper are highly synergistic and complementary, as the second set of analyses could not have been completed without the identification and characterization of the acute response genes identified in the first part. In the revised manuscript, we have revised the text thoroughly to highlight this synergy and create more cohesion across the analyses and results presented.

Overall issues.

The title refers to two modules of HIF1A network, but these are not described other than in passing. Are these related to acute versus secondary response gene cohorts based on pro-seq or as oncogenic versus tumor-suppressive genes defined by database scouting? Is knowing acute response genes potentially diagnostic for cancer outcomes or not? If acute response genes do not define outcomes one way or the other, then part 1 of the manuscript does not add much toward the second part and hence my harping about the manuscript seeming to be in two pieces. This issue appears crucial to me for the

manuscript's overall impact. It would be great if it can be addressed by an argument and I apologize in advance if I may have overlooked this in the existing text.

Response: We thank the Reviewer again for the constructive feedback and the opportunity to better explain the importance of the results in the manuscript and how the various results and discoveries are integrated. The two modules referenced are the **tumor suppressive** and **oncogenic** modules within the **acute transcriptional response**. These modules do not refer to 'direct' versus 'indirect' hypoxia-inducible genes. As explained in our response above, the identification of these two modules with divergent, context-dependent roles in cancer biology could not have been possible without first defining and characterizing the acute response to hypoxia with direct measurements of transcriptional changes. Therefore, the first part of the manuscript enables the second part of the manuscript. By defining first the acute transcriptional response to hypoxia and its strong dependency on HIF1A, we were then able to analyze the cancer relevance of acute response genes. Knowing the acute genes enables a more precise analysis of the direct hypoxia response (as opposed to downstream indirect pathways) in cancer biology. These efforts revealed that the direct acute hypoxic response involves genes with both tumor suppressive and tumor promoting properties, whose action is revealed in different contexts (i.e. DDIT4 as suppressor of cell viability in normoxic, nutrient-rich conditions versus ECM remodeling enzymes as oncogenes in advanced human tumors). In the revised manuscript, we have revised the text to further integrate the results and explain the flow of activities, from gene discovery first, to analysis of cancer relevance next.

Is there a reason to believe that rapid and transient activation of acute response genes under a hypoxia pulse involves the same mechanisms and, furthermore, illuminates the same network modules as stable steady-state overexpression of these targets observed in patient samples? There may be a generalization here about mechanistically connecting acute responses and long-term expression changes that is implied, but not tested or at least discussed.

Response: Thanks for the opportunity to elaborate on this point, which relates directly to our operational definition of acute response genes. As explained in the manuscript, acute response genes are both transactivated upon acute hypoxia **AND** their mRNA levels are significantly elevated in steady-state RNA measurements at 24 hours hypoxia. We also identified some genes that were transactivated in the acute PRO-seq experiment but not so in the steady-state measurement at the later time point by RNAseq, but these genes were not included in the acute response gene list and therefore not used for downstream analyses of CRISPR genetic screen data or data from patient samples. With that being said, it is likely that hypoxia induces late and/or indirect responses that also play roles in cancer biology, but those late responses are not analyzed in our study.

Our choice of focusing our analysis on **acute response genes** is based on three important observations. First, we found that both the absolute transcriptional output and the transcriptional fold change among acute response genes have a direct correlate in the steady-state mRNA levels at late time points. Genes strongly transcribed and strongly transactivated during the immediate early response are expressed at the highest levels and induced more strongly at the mRNA level later on. Second, we report in the manuscript that whereas acute response genes are largely conserved across diverse cancer cell types, late response genes are not. Lastly, the acute response genes are associated with strong and conserved HIF1A binding events. The revised manuscript explains our choice to focus our analysis of cancer relevance only for acute response genes, while acknowledging that late response genes could also play important roles in cancer biology.

As presented, I am not fully clear about how CDK8 fits in this story. CDK8 is supposed to be globally involved in transcription as part of Mediator: does connection of CDK8 with HIF targets reflect a special relationship between CDK8 with HIF or CDK8 requirement/ involvement extends to activation in general including any other acute response? This issue might be addressed by discussion.

Response: Thanks for the constructive criticism, which prompted us to revise the discussion of the CDK8 results. The Reviewer correctly points out that CDK8 is a component of the Mediator complex, and that the Mediator complex is considered a global regulator of transcription. However, CDK8 is not an obligate subunit of the Mediator complex, and it interacts with Mediator in a mutually exclusive fashion with its close paralog CDK19⁹. Therefore, whereas core Mediator subunits may have more global roles in transcription, CDK8 and CDK19 have clearly specialized roles in largely non-overlapping transcriptional programs¹⁰.

CDK8 has been implicated in both activation and repression of transcription¹¹. To date positive effects of CDK8 in transcription appear to be limited to select stimulus-responsive transcription networks, of which the HIF1A-dependent transcriptional network is one^{10,12}. Previous work by us and others has demonstrated that CDK8 and its close paralog CDK19 play specialized roles in different transcriptional networks. In the hypoxia-inducible program specifically, CDK8 plays a major role, but CDK19 does not¹⁰. Of key relevance to the new results shown in our manuscript, we and others have shown that CDK8 and CDK19 can exert kinase-independent roles in some contexts¹³⁻¹⁵. Therefore, we believe it was important to formally demonstrate a role for CDK8 **kinase activity** using the chemical genetics approach described in the manuscript. Given the strong interest in development of specific CDK8 inhibitors for therapeutic purposes¹⁶⁻²⁵, we consider these results to be relevant and worthy of inclusion in the manuscript.

Is this unusual that some genes appear inhibited by CDK8? Is this a direct acute effect (and if so, through what mechanisms) or a consequence of incomplete inhibition wherein the remaining CDK8 is delivered to certain genes? This is related to the statement about genes requiring CDK8 for transactivation. Also, on the same page, what is meant by “significant impact on productive transcription and pausing indexes” needs to be spelled out: increase, decrease, etc?

Response: It is not unusual to see evidence of CDK8-dependent gene repression. In fact, CDK8 was first characterized as a repressor of transcription by a number of mechanisms, including kinase-dependent inactivation of the general transcription factor TFIID²⁶, kinase-independent blocking of Mediator interaction with the Pre-Initiation Complex (PIC)¹⁴, phosphorylation-dependent turnover of sequence-specific transcription factors²⁷, and kinase-dependent attenuation of ‘super-enhancers’²⁸. Our team was among the first to demonstrate a positive role for CDK8 in gene activation through stimulation of RNAPII elongation^{10,29}, but these earlier studies did not differentiate between kinase-dependent and -independent roles. Therefore, CDK8 plays both positive and negative roles in control of transcription, with kinase-dependent and -independent roles, which increases the importance of our findings demonstrating **widespread kinase-dependent roles within the acute hypoxia response**. We have clarified the sentence pointed out by the Reviewer to state: ‘Among hypoxia-inducible genes that require CDK8 kinase activity for transactivation, CDK8 inhibition significantly reduced productive elongation without strongly affecting transcription at TSS, reinforcing the notion of HIF1A and CDK8 working coordinately to stimulate RNAPII pause release’.

Specific questions.

Pro-seqs are not possible to interpret as it is unclear what signal looks like between individual biological replicates. As a method highly prone to variability in reported pausing indexes due to different enzymes involved in generation of clonable promoter (decapping) versus downstream (PNK) RNAs, showing individual replicates +/- hypoxia is essential. This is especially due to coverage (<<100M reads) being modest by pro-seq standards. This also applies to supplemental table 710004 where base values should be shown for each replicate.

Response: In response to this comment, we would like to emphasize that the statistical analysis of PRO-seq results performed with DESeq2 takes into account the variability across replicates, while also accounting for the fact that lowly transcribed regions display more variability³⁰. Nevertheless, to further emphasize this point and highlight the reproducibility of our PRO-seq results, the revised manuscript includes now a display of reproducibility across replicates in the PRO-seq experiments in new panels in

Supplementary Fig. 1b-c and Supplementary Fig. 4a-b, and the independent replicate values were listed separately in Supplementary Table 1, (now tabs h and i).

Would it be possible to show how pausing changes on repressed genes using a metaplot and show an example of a downregulated gene as a browser shot?

Response: Thanks for this good suggestion. The revised manuscript now includes a comparison with upregulated genes showing that pausing index actually increases upon hypoxia at repressed genes (new Supplementary Fig. 2a) and a metaplot for repressed genes (new Supplementary Fig. 2b). We also now include browser shots of example downregulated (CRELD2) and unchanged (UBE2B) genes in new Supplementary Fig. 1e.

Given an apparent similarity in repression between wild type and Hif^{-/-} cell types, a graph similar to 1g, but with repressed genes should be shown as well.

Response: Thanks for this suggestion. The requested graph is included in Supplementary Fig. 2b.

Figure 1a (browser shot) needs a negative control locus that does not change with hypoxia. A repressed gene example should also be shown either in main or supplement.

Response: We have included these two examples (i.e. unchanged and repressed) in Supplementary Fig. 1e.

Figure 1d. Is it surprising to see that apart from a handful of highly activated genes, there is overall a pretty good positive correlation between pro-seq gene body signal changes in wild type versus ^{-/-} cells? Does this mean that Hif1a controls only a small part of hypoxia response and the rest of the response occurs without Hif?

Response: In response to this comment, we would like to clarify that Figure 1d displays **the entire transcriptome**, not just hypoxia-inducible genes. Genes *significantly* affected by hypoxia are highlighted in red (upregulated) and green (downregulated) and are clearly deviated from the diagonal (i.e. greater fold-change in WT than in HIF1A^{-/-} cells), whereas the majority (highest density) of genes are clustered around 0 on both axes (no change in either cell line), with a smaller fraction around the diagonal (i.e. no difference between cell lines).

Figure 1i is confusing and should be shown as actual base values before and after hypoxia. This will allow one to better judge relative levels of basal expression of these genes.

Response: Thanks for this constructive comment, which prompted us to display these results as RPKM Z-scores for each gene across the various genotypes and treatments for greater clarity. The actual RPKM values can be found in Supplementary Table 1. Displaying the Z scores helps prevent any one gene/condition from dominating the color scale and highlights the differences in basal expression in normoxic conditions.

CDK8 section. Having pro-seq with an inhibitor-sensitive mutant is appreciated on its own. Is there a difference in basal transcription and/or pausing indexes between wild type and mutant cells without the inhibitor?

Response: Thanks for this question, which prompted us to elaborate on this topic in the revised manuscript. The inhibitor-sensitive CDK8 mutant behaves as a hypomorph, in the sense that it shows partially reduced kinase activity even before adding the bulky ATP analog³¹. Therefore, even in the DMSO condition hypoxia-inducible genes show decreased expression, but the requirement for CDK8 kinase activity is more fully evident when cells are treated with the ATP analog. To elaborate on this point, in the revised manuscript we show both the effect of complete CDK8 inhibition in main Figure 3

(+3MB-PP1 conditions) and the partial effect in the DMSO condition in new panels in **Supplementary Fig. 4**.

Line 232. What is “highly” referring to when talking about commonality of acute response genes across cell lines?

Response: Thanks for this comment. We have revised this sentence for clarity as follows: ‘Overall, these results reveal that acute response genes are more conserved across cancer cell types than late response genes, commonly associated with high-occupancy HIF1A binding sites, and thus more likely to play a role in the cellular adaptation to hypoxia across multiple cancer types’.

Discussion. Is the first paragraph of the discussion necessary? Parts of it may belong to introduction. The discussion could be shortened as well.

Response: Thanks for this comment, which we address in the revised manuscript with a more streamlined discussion.

Figure S2c. Could authors speculate as to why some downregulated genes have Hif1A peaks at promoters (if I am understanding this graph correctly)?

Response: Thanks for this comment, which led us to elaborate further about this observation in the revised manuscript. Our data shows that there is a genome-wide ‘baseline’ level of HIF1A binding at hundreds of TSSs across the genome upon hypoxia, and that this baseline binding is not significantly different from unchanged genes (not significant, n.s.) at downregulated genes. In contrast, upregulated genes show significantly elevated HIF1A binding at TSSs. This baseline binding at TSSs could be explained by the fact that the hypoxia response element (HRE) is rather short and likely to occur stochastically at many positions across the genome, and that HIF may access these HREs at nucleosome-depleted regions such as TSSs. We believe this observation is an important contribution from our work, in the sense that previous studies have simply used combined lists of HIF1A peaks versus mRNAs showing changes at steady-state level at late time points to identify ‘HIF1A targets’, an exercise that is likely to produce many ‘false positives’, as HIF1A binds to a large number of genomic regions, not all of which are directly transactivated by it. Therefore, by using measurements of RNA synthesis at short time points, we have produced a superior list of direct HIF1A targets.

Figure S2f. How were the attributions between distant peaks and genes done for distal HIF1 peaks?

Response: Peaks were called by the Homer software and linked to genes solely by distance from TSSs (limited to distances of **5-50 kbp** for distal peaks).

Reviewer #2 (Remarks to the Author):

In this manuscript, Andrysiak et al. have provided a deeper understanding of HIF1A mediated changes in transcription during hypoxia, and have also identified contextual roles of HIF1A in regulating cancer. To identify acute response of hypoxic stress they have measured nascent RNA transcription by PRO-seq and have identified hundreds of genes to be up- and down-regulated in a HIF1A dependent manner. They have demonstrated that activated genes have increased HIF1A binding relative to the repressed ones. Such activation by HIF1A is mediated by the release of paused Pol II and requires CDK8 kinase activity. To distinguish between acute vs late response genes they have compared the early time-point PRO-seq data with steady-state RNA-seq after 24 h of hypoxic stress, showing that the transcription of acute response genes is significantly correlated with HIF1A binding and active chromatin marks. The authors find that most of the acute response genes in HCT116 are activated across other cancer cell types and the core genes show higher HIF1A binding. They further analyzed the genome-wide knock-out data in hundreds of cancer lines, and identified a tumor suppressive role of HIF1A that could be orchestrated by DDIT4 mediated repression of the mTOR signaling process in normoxic

conditions. Interestingly, the authors also identified that high-expression of the acute response genes were associated with decreased survival in multiple cancer types, suggesting that HIF1A has contrasting roles as it could act as a tumor suppressor or as a mediator of cancer progression. In summary, the authors have done an excellent job in dissecting an important biological phenomenon, and have well-utilized a high-resolution assay to probe nascent transcription in combination with ChIP-seq and other available datasets that have provided deeper insights into HIF1A dependent transcription and its roles in hypoxia and cancer. The manuscript is very clearly written and experiments are properly executed. I believe it will be a significant contribution to the field and serve as a resource for future studies. Here are few comments that could improve the manuscript further.

1. The authors have performed PRO-seq after 90 min of hypoxic stress to identify early responsive genes but have looked at HIF1A binding after 24 h of stress. As the levels of HIF1A look similar at 90 min vs 24 h, I wonder why the authors did not do the ChIP-seq after 90 min as well. Minor point: There is a typo in Fig. 1a (Normoxia should be 21%).

Response: Thanks for this comment, which prompted us to better explain in the revised manuscript the choice of the time point for HIF1A ChIP-seq. As noted by the Reviewer, total cellular levels of HIF1A measured by western Blot are similar at 90 minutes and later time points. However, given that we performed PRO-seq in only one of the cell lines (HCT116) and RNAseq at the later time point of 24 hours in all four cell lines (HCT116, RKO, A549, H460), we chose to match the RNAseq and the ChIP-seq time points for a more valid cross-cell line comparison. Furthermore, recent work by Smythies et al³² showed that HIF1A chromatin binding patterns are relatively unaffected by either time point of hypoxia exposure or exact O₂ levels used to stabilize HIF1A.

2. Minor point: In line 83 the authors say that Pol II pauses at ~50-100 nt. Actually, this estimate is around 20-60 bp from the TSS (Tome et al. Nat Genet. 2018).

Response: Thanks for this comment, which we address in the revised manuscript, including citation of the manuscript brought to our attention by the Reviewer.

3. For Fig. 1c it might be helpful for the reader to see some kind of a significance cut-off that determines if the genes that have altered PRO-seq signal in the HIF1A deletion line are really significant or not. Most of the changed genes in the mutant are lowly significant relative to the WT, so are they really changed? Looks like without HIF1A there is no significant alteration of transcription. It would be also useful to quantitatively distinguish the HIF1A-dependent and the -independent genes that are altered by hypoxia and check the HIF1A binding characteristics (as done in Fig.2) of the HIF1A-dependent genes. Minor point: It will be nice to see the replicate correlation plots for the PRO-seq data in the supplement.

Response: Thanks for these comments, which we address in the revised manuscript. In the revised **Figure 1c**, we have added a horizontal dashed line in the Volcano plots to help readers identify the statistical cut off of 10% FDR used in our DESeq2 analysis. Please note that numbers of significant genes (FDR10) in each cell line are indicated above the plots. We also clarify in the legend that all colored dots in these plots indicate *significant* changes, including those observed in the HIF1A^{-/-} cell lines. We also mention in the text that the 'residual' significant changes observed in HIF1A^{-/-} cells could be potentially driven by HIF2A. In response to the Reviewer's comment, we have added 'meta plots' for HIF1A binding at HIF1A-dependent and -independent genes in **Supplementary Fig. 3d**. Lastly, we have added a display of the PRO-seq replicates (also requested by Reviewer 1) in **Supplementary Figs. 1b-c and 4a-b**.

4. It will be useful to see the global gene-body and the TSS correlations of the WT and the mutant lines to validate that global transcription is not altered upon HIF1A deletion. Also, how does the metagene plot look like for unregulated genes in WT and mutant lines in the two conditions?

Response: Thanks for these suggestions, which we address in the revised manuscript with scatter plots and metagenes. Simply put, these analyses reveal that there are no significant global differences in the transcriptome of HIF1A^{-/-} cells. Scatter plots for gene-body and TSS correlations are shown in

Supplementary Figs. 2d-e. Metagene for unchanged genes is shown in **Supplementary Fig. 2c.**

5. Interestingly, the authors observe a modest increase of paused Pol II in the WT cells upon hypoxia at the upregulated genes (Fig.1g). Does this suggest that Pol II recruitment increases at the upregulated genes in a HIF1A dependent manner? Does the pause signal also go up in the downregulated genes?

Response: Thanks for this comment. Indeed, there is an increase in PRO-seq signals at TSSs upon hypoxia, although more modest than at gene bodies. We also include now an analysis of pausing at downregulated genes, which shows that TSS signals decrease mildly but significantly at repressed genes, along with an increase in the pausing index (new **Supplementary Fig. 2a**). Thus, whereas the bulk of transcriptional changes at hypoxia-inducible and hypoxia-repressed genes occurs at gene bodies, there is still an impact of hypoxic signaling (albeit to a lesser quantitative degree) at TSSs.

6. Did the authors analyze transcriptional changes in the paused TSSs at intergenic regions to explore the idea that HIF1A could be necessary for enhancer transcription? In this context, the authors mention in the introduction that the acute response of core genes are associated with strong HIF1A enhancers. However, I did not see any distal transcriptional regulatory element/enhancer specific analysis. Are they referring to distal HIF1A binding sites as enhancers? Do these sites show divergent transcription upon hypoxia and other enhancer specific histone marks? Minor point: The authors show non-coding RNA transcription in Fig. S3 but refer to it as S4 in the text. Please check.

Response: Thanks for this suggestion, which prompted us to include and discuss new results about the interplay between HIF1A binding, bidirectional transcription at distal sites, and enhancer chromatin marks. In the revised manuscript, we show that, in agreement with what has been observed for other transcription factors, distal HIF1A binding sites associated with hypoxia inducible genes display bidirectional transcription, as well as histone marks of enhancers, such as increased DNase I accessibility, H3K4me1, and H3K27ac. These data are shown in new **Figure 2 panels d-f**. Notably, we completed a comparative analysis of distal (i.e. up to 50kb from TSS) HIF1A binding events associated with nearby transactivation events (i.e. productive binding) versus distal binding events not associated with transcriptional changes in nearby genes (i.e. non-productive binding). This comparison reveals that productive HIF1A binding at these distal elements is associated with eRNA production and presence of enhancer histone marks even in normoxia, before HIF1A stabilization. Thus, although HIF1A binds to thousands of distal genomic regions, only the binding at those regions that are being transcribed in normoxia and harbor chromatin marks of enhancers will lead to hypoxia-inducible changes in nearby genes. These results are consistent with the notion of HIF1A acting preferentially at sites of pre-formed enhancer-promoter contacts.

7. The authors have defined gene body as +1001 to end of the gene. What is the end of the gene? How far is it from the cleavage and polyadenylation site?

Response: This was previously explained in the Methods section. We further clarify now in the Methods that 'the end of the gene' is defined by the cleavage/polyadenylation site corresponding to the shortest version of the annotated gene according to RefSeq.

8. For the CDK8 analog sensitive studies I see that the authors used a DMSO control for the CDK8 as/as cell, but did not show a metagene plot for that control with respect to the WT/treated lines in the normoxic and hypoxic conditions. It will be useful to see this comparison. In line 671 did the authors mean WT, mutant and CDK8 as/as HCT116 cells? Please check. It looks like the authors did a PRO-seq with WT and mutant lines in the norm and hypoxic conditions, and then did another PRO-seq with the WT and the CDK8 as/as line in presence of DMSO or 3MB-PP1 in the above conditions. This design is not clear in the methods section (line 671).

Response: We welcome this comment, also raised by Reviewer 1, which we address in the revised manuscript with a better description of the experiment and additional data displays. As explained above in response to Reviewer 1, we performed PRO-seq in the CDK8as/as and WT parental lines, in

normoxia and hypoxia, and with and without the ATP analog inhibitor. The CDK8as/as line behaves as a hypomorph, with partial loss of CDK8 kinase activity in the absence of the analog, and complete inhibition in presence of the analog³¹. The impact on hypoxia-inducible genes is therefore 'graded', with partial inhibition in the DMSO vehicle control, and more profound inhibition upon addition of 3MB-PP1. We clarify this by showing more clearly labelling the results from the +3MB-PP1 conditions in revised **Figure 3**, and now showing the 'DMSO' conditions in new **Supplementary Fig. 4**.

9. The authors claim that hypoxia induces release of TSS-proximal Pol II mediated by CDK8. However, from Fig. 3f, it appears that the TSS Pol II levels are lower in the CDK8as/as treated cells relative to WT in the hypoxic condition, suggesting that CDK8 is essential for hypoxia driven Pol II recruitment/initiation as well. Please explain.

Response: Thanks for this comment, which we address in the manuscript with the discussion of changes in TSS, gene-body and 'pausing indices', which take into account the relative impact on TSS versus gene body. Simply put, CDK8 inhibition affects signals at both TSS and gene bodies, but the effect is quantitatively greater at gene bodies. This quantitative difference is evident in **Figure 3g**. Because of this differential between TSS and gene-bodies, the significant decrease in pausing indices observed in WT cells is no longer evident in the CDK8as/as lines (**Figure 3g**).

Reviewer #3 (Remarks to the Author):

In this study, the authors examine HIF1A-mediated gene transcription in cancer cell lines. Using Precision Run-On with sequencing (PRO-seq), which method only detects transcriptionally-engaged RNA polymerases, HCT116 cells were profiled for HIF1A-mediated transcription at 90 minutes of hypoxia conditions. The Pro-seq results were compared with conventional RNA seq data from multiple cell lines under hypoxia conditions for 24 hrs, where many of these 24hr changes would involve indirect effects as compared to the acute and direct changes as measured in the Pro-seq data. Chip-seq data of HCT116 and RKO defined bound targets of HIF1A. HIF1A ChIP-seq data indicates that hypoxia-driven transactivation, but not repression, is a direct effect of HIF1A. Analysis of publicly available genetic screen data produced by the DepMap Project revealed HIF1A and genes in a HIF1A network to be essential in normoxic cells. Analysis of TCGA expression and survival data showed that higher expression of acute response genes as a group is more commonly associated with shorter survival across different cancer types.

Specific comments:

1) It seems that the unique advantages of the PRO-seq platform (base-pair resolution and strand-specific information, rare and common nascent RNAs, unstable nascent RNAs transcribed from enhancer regions) are not really being utilized in this study. Basically, the data are collapsed into the known genes, where there have been previous and numerous expression profiling studies of hypoxia and of defining HIF1A-specific targets. The PRO-seq method is unique from the approach of other studies, but are the key advantages specific to PRO-seq able to yield new insights into HIF1A?

Response: We are grateful for this comment, which prompted us to better explain the key advantages of our study relative to previous studies of the transcriptional response to hypoxia. The Reviewer is correct in pointing out that our manuscript is focused mostly on known genes, with lesser focus on transcriptional events elsewhere in the genome. Nevertheless, following Reviewer's guidance, the revised manuscript includes now an analysis of bidirectional transcription at distal sites (see response to Reviewer 2 point 6 above) and a direct comparison of our PRO-seq approach versus the more canonical approach of cross-referencing HIFA ChIP-seq data with steady state mRNA measurements such as RNAseq (see new **Supplementary Fig. 5a-b**).

First, our study enables a better dissection of true acute/early HIF1A-dependent transcriptional responses versus late, indirect, or post-transcriptional responses. As explained above in response to

Reviewer 1 comments, previous studies have employed a combination of HIF1A chromatin binding measurements and mRNA steady-state measurements to ascertain 'HIF1A targets'. As demonstrated in our manuscript, these approaches, although valuable, are likely to produce many 'false positives', because they would ascribe 'direct HIF1A transactivation' to genes that are actually regulated *indirectly* by HIF1A, either through secondary transcription factors or even post-transcriptional stabilization. As highlighted in **Supplementary Fig. 3g-h**, ~11% of the genes in the human genome display HIF1A binding near TSSs, and ~15% display binding within 5-50kb. Although this binding is real, and likely mediated by stochastic occurrence of HREs in open chromatin regions, our study shows that most of this binding is inconsequential in terms of direct transactivation by HIF1A. As shown in new **Supplementary Fig. 5a-b**, there are hundreds of 'late response genes' with nearby HIF1A binding, but at the present moment it is impossible to define whether they are direct or indirect targets of HIF1A without embarking on high-throughput genome editing of the associated HREs. The new **Figure 4** and **Supplementary Fig. 5** directly address the question by the reviewer about what is novel when using PRO-seq. We copy here the corresponding fragment of the text:

*'This exercise reveals that simply cross-referencing steady-state mRNA data with HIF1A ChIP-seq data would predict the existence of hundreds of putative direct HIF1A targets that are not immediately induced by hypoxia (i.e. 457 genes when using proximal + distal sites, **Supplementary Fig. 5b**). Additionally, this approach would miss dozens of acute response genes identified by PRO-seq for which nearby HIF1A is not evident, including many novel hypoxia-inducible, HIF1A-dependent genes'.*

Second, by identifying the acute response to hypoxia, which is largely HIF1A-dependent, we were able to complete a suite of downstream analysis, such as investigation of chromatin features, CDK8 dependency, or roles in cancer biology, with the confidence that our gene list truly represents the direct transcriptional response to hypoxia. As demonstrated in **Figure 5**, the acute response genes defined by PRO-seq are much more conserved across cancer cell types relative to all other hypoxia-inducible mRNAs, and therefore more likely to mediate biological responses to hypoxia in a variety of settings.

We appreciate the challenge by the Reviewer to better explain the advantages of our approach and new insights obtained, which we do in the revised text and new figure panels in response to comment #2 below.

2) The PRO-seq results at 90 minutes of hypoxia are compared with RNA-seq at 24 hr. The 24hr profiles are going to have more genes differentially expressed, due to the indirect effects and longer time frame. Are the Pro-seq data finding acute and direct targets that would not have been uncovered by conventional RNA-seq? For example, if conventional RNA-seq at 90 minutes is carried out, would additional indirect effects be found at that time point? What if the RNA-seq results are overlapped with the Chip-seq results; would this filter out the indirect effects in a similar manner to using PRO-seq? Here, we would not be as interested in the total RNA levels as we would with the differential changes in response to hypoxia.

Response: We welcome this comment, which is related to the comment #1 above, and which we address in the revised manuscript with additional analysis of the PRO-seq, ChIP-seq, and RNA-seq datasets. The Reviewer is absolutely correct that the 24 hours RNA-seq datasets will have many more genes due to indirect effects, which would confound the identification of direct versus indirect targets, even when filtering by nearby ChIP-seq signals. In the new **Figure 4** and **Supplementary Fig. 5**, we address this point directly by asking: how many of the hypoxia-inducible mRNAs at 24 hours have proximal (or distal) HIF1A binding and/or increased signals at 90 minutes by PRO-seq? This exercise reveals a large number of hypoxia-inducible mRNAs arising from genes that harbor nearby HIF1A peaks but whose transcription was not elevated by PRO-seq at the early time point. Therefore, these genes would have been annotated as direct HIF1A targets by the commonly used 'RNAseq-filtered-by-ChIPseq'. This exercise reveals the value of the PRO-seq dataset, as it enables the field to focus on a 'high confidence' list of direct targets transactivated at early time points of hypoxia.

3) Page 7 notes that "the acute transcriptional response identified by PRO-seq also includes many genes that have not previously been linked to hypoxic signaling". Are we sure about this point? There

have been many previous profiling studies of hypoxia, a number of which (Winter, Buffa, Ragnum) were recently examined in the PCAWG hypoxia paper (<https://www.nature.com/articles/s41467-019-14052-x>). It may be that many genes in the current study have not been highlighted or focused on in previous studies, but they may be found in the previous expression profiling results. If one takes the PRO-seq hypoxia genes and compares them side-by-side with results from a number of different datasets available on GEO (in a manner similar to the Figure 5b representation), does a signature truly unique to the PRO-seq data emerge?

Response: In the original manuscript, in **Supplementary Table S2**, we provided an exhaustive analysis of the PRO-seq gene data set versus 21 published studies aimed at identifying HIF targets, including some of the papers mentioned by the Reviewer, indicating which genes had been identified in which studies. This effort revealed that nearly half of the PRO-seq genes (~400) had been reported in at least one other study, with the other half being absent from those datasets. Prompted by the Reviewer's comment, we have updated this table with a several additional studies (all the studies mentioned by the Reviewer are included in this analysis). Whereas some hypoxia inducible genes have been reported as such in >10 studies, dozens of them are reported here for the first time (to the best of our knowledge).

4) There are have been many previous studies of HIF1A binding by Chip-seq (e.g., lots of data available on GEO), though perhaps not in the HCT116 cell line. Do the observed cell type specific differences in HIF1A binding represent a novel finding or has this been previously observed?

Response: We welcome the invitation to better place our discoveries in the context of previous studies of HIF1A chromatin binding. The Reviewer is correct that there are many previously reported ChIP-seq experiments, yet our study is the first to couple ChIP-seq with nascent RNA measurements. Strong cell type-specificity in the gene expression profiles evoked by hypoxia have been previously documented and discussed (reviewed by Schodel et al³³), but the mechanisms driving this diversity await elucidation. Side-by-side comparison of HIF1A ChIP-seq signals and 'Capture C' signals in two different cell lines by Platt et al³⁴ documented cell type-specificity in HIF1A binding, which was explained in part by variations in 'pre-formed', HIF1A-independent patterns of chromatin conformation. Within this context, the value added by our side-by-side ChIP analyses of four different cell types is two-fold: 1) While we confirm the previous notion of high diversity in HIF1A chromatin binding patterns, our quantitative assessment demonstrates that this diversity applies mostly to 'low occupancy' HIF1A binding sites (**Figure 5e-g**); and 2) Acute response genes identified by PRO-seq are clearly associated with strong, conserved HIF1A binding events (**Figure 5f-g**). Prompted by the Reviewer comment, we revised the text to better explain what is new in our study relatively to what was previously known.

5) Do the CDK8-related findings of the present study yield new insight into CDK8 and HIF1A, beyond previous studies such as the cited Galbraith, M. D. et al. study in Cell?

Response: Thanks for this comment, which we addressed in part in our responses to Reviewer 1 above. As noted by the Reviewer, we previously reported that CDK8 plays a specialized role in the hypoxia network as a widespread positive regulator of HIF1A target genes¹⁰. However, our previous study did not define whether CDK8 exerts these effects via kinase-dependent or -independent effects. Given the increasing appreciation for kinase-independent functions for both CDK8 and CDK19 in transcriptional control¹³⁻¹⁵, and the strong interest in the development of CDK8 inhibitors for therapeutic purposes¹⁶⁻²⁵, we believe our new data demonstrating a kinase-dependent role for CDK8 in HIF1A-dependent direct transactivation is a valuable addition to this manuscript specially and to the field more generally. In the revised text, we make this distinction between the previous work and the new data more explicit.

6) Using histone data from ENCODE, the authors find that HIF1A drives its core program via high-occupancy enhancers within open chromatin. Is this something that would be unique to HIF1A as compared to other transcription factors? Or would this represent new insight from previous studies such as the cited Platt, et al. EMBO Rep paper?

Response: With regards to the generality of our finding on HIF1A relative to other transcription factors, it is unclear at this point whether the existence of 'core programs' associated with strong enhancers at sites of open chromatin would be a prevalent phenomenon. In our studies of the p53 transcriptional program in different cell types, we made a similar observation (i.e. core p53 target genes tend to bind more p53)³⁵, so we don't think this would be unique to HIF1A. However, this observation is unlikely to apply to all transcription factors, specially the so called 'pioneer' transcription factors known to be able to access their enhancers even in the context of closed chromatin³⁶. Regarding the value added relative to the work by Platt et al, their study focused on the role of a *limited number* of 'pre-formed' chromatin contacts in defining responses to HIF1A, which is supported by our analysis of ENCODE data, but they did not assess quantitative differences in HIF1A binding between common and cell type-specific genes. Another new insight in our revised manuscript, which was prompted by a reviewer comment, is that 'productive' distal HIF1A binding associated with transcriptional changes at nearby genes is associated with bidirectional transcription (i.e. eRNAs). In the revised text, we discuss these observations in more detail.

7) The use of the hypergeometric test (Figure S2, Figure 4a) is most assuredly incorrect. The hypergeometric test evaluates the chance of getting the EXACT number of overlapping genes found. The correct test would be one-sided Fisher's exact or chi-squared, which evaluates the chance of getting the observed number of overlapping genes OR MORE.

Response: Thanks for the comment. The hypergeometric tests in the original manuscript were carried out in such a way as to estimate the probability of observing a *given overlap or greater*. Nonetheless, in the revised manuscript, we now report the p values and odds ratios obtained from two-sided Fisher's exact tests (which allows testing for both under and over-representation in the overlap of gene sets simultaneously).

8) Figure 4d compares Pro-Seq RPKM with conventional RNA-seq RPKM. It would seem that comparing differential levels (compared to normoxia) would be more appropriate to see if the HIF1A-associated changes are sustained.

9) Similar to the above point, for Figure 4f, would using differential expression (vs normoxia) instead of gene body RPKM be more relevant?

Response: Thanks for these comments, which we address by adding the comparison of 'transcriptional output' versus 'fold changes' in new **Supplementary Figs. 5f-g and 6a-b**. Whereas both the early transcriptional output and degree of transactivation (i.e. fold change) have a direct correlate in the late steady-state transcriptome, only the transcriptional output is positively associated with marks of open active chromatin. In contrast, the strength of HIF1A binding is positively associated with both transcriptional output and fold change. In other words, HIF1A binding to its 'cistrome' could impact on how much a gene is transcribed and how much it is induced upon hypoxia, but the chromatin environment seems to be the key determinant of the absolute transcriptional output from each target loci and therefore the relative stoichiometry of hypoxia-inducible mRNAs.

10) The DepMap results would seem to stand on their own but be somewhat disconnected from the other analyses in the paper. Would there be any enrichment or global association between the HIF1A acute targets and the DepMap-related HIF1A network?

Response: We agree with the Reviewer that the DepMap analysis represents a transition point in the manuscript, where we pivot from identification of the acute response program genes to analysis of their potential contributions to the context-specific tumor suppressive and oncogenic roles of HIF1A. The answer to the Reviewer's question is NO (**Figure 6b**), there is no obvious global association between acute targets and the DepMap-related HIF1A network. Acute targets reside all over the 'DepMap' rank (red dots **Figure 6b**). However, some key acute targets (e.g. DDIT4, FOXO3) rank among the most strongly correlated with HIF1A in this 'genetic interactome'. In other words, very few acute response

genes mimic HIF1A in terms of its anti-proliferative effects in the context of nutrient rich normoxic conditions. This is an important result, showing that the acute response program is clearly multifunctional, with HIF1A-mediated suppression of mTOR signaling being the key activity in this context.

11) For the TCGA survival analysis, the endpoint is described in the text and figures as “Progression-Free Survival”. This would seem entirely incorrect. It seems more likely that Overall Survival was the endpoint. There is a big difference between the two endpoints. Most cancer types in TCGA don't have progression free survival data available.

Response: Thanks for this comment, which prompted us to better explain our choice of PFS (now referred to as progression-free interval (PFI) in the revised manuscript to better distinguish from overall survival) as the key endpoint. Whereas Overall Survival (OS) is an important endpoint with minimal ambiguity, using OS as an endpoint requires sufficient follow-up time that depends on the aggressiveness of the disease, minimum follow-up times for PFI are shorter as progression events tend to occur earlier than death. Major limitations of the original TCGA data are the lack of comprehensive clinical outcome data and the generally short follow-up times after initial tumor sampling. However, a group working with The Cancer Genome Atlas Research Network recently released a curated, standardized clinical dataset, the TCGA Pan-Cancer Clinical Data Resource (TCGA-CDR)³⁷ with derivation of multiple clinical endpoints, including PFI and OS for many cancer types, along with cancer-specific recommendations for endpoint use. Due to the longer follow-up times required for OS and based on these recommendations, we chose to use PFI as the endpoint in our analysis of TCGA expression data. Nevertheless, in response to the Reviewer's comment, we repeated the analysis using OS as the endpoint, which produced very similar results. The key result showing that hypoxia-inducible ECM remodeling genes are consistently associated with poor prognosis is evident when using either endpoint, PFI or OS. The results of the OS analysis are now summarized in **Figure S9e**.

12) For all KM plots, the numbers of patients in each arm needs to be indicated. Also, why were different splits used for different datasets?

Response: Thanks for this comment, which we address in the revised **Figure 6** by indicating the numbers in each arm. More comprehensive information is included in **Supplementary Table 7**. Regarding the different splits, we used an iterative KM log-rank approach to find the optimal split between samples with high and low scores (or individual gene expression), as detailed in our Methods section. We have added a statement to clarify this in the main text and refer readers to the Methods section.

13) A major limitation of using TCGA data for the survival analysis is that for most cancer types the survival data are not mature. For a few cancer types like kidney one can identify robust survival correlates, but for other cancer types such as lung or breast the patients have not been followed up for long enough. This would mean that many of the negative results could be the result of insufficient data rather than a true lack of association. It would be useful to also check additional public datasets (e.g. METABRIC) for which the survival data are adequate.

Response: As noted above, we chose to use PFI in our analysis of TCGA data due to the generally shorter follow-up times required to obtain sufficient data. Furthermore, we excluded from our analysis cancer types for which use of PFI was not recommended in the TCGA-CDR and any comparisons in which either arm had $n < 10$. Following Reviewer's guidance, we added text in the revised manuscript to acknowledge that some cancer types may not have sufficient follow up time, encouraging cautious interpretation of negative results.

14) For the figure legends, whenever a p-value is represented in the figures, the specific test used to derive the p-value should be noted.

Response: Thanks for pointing this oversight, which we have corrected in the revised manuscript.

References.

- 1 Batie, M. *et al.* Hypoxia induces rapid changes to histone methylation and reprograms chromatin. *Science* **363**, 1222-1226, doi:10.1126/science.aau5870 (2019).
- 2 Chakraborty, A. A. *et al.* Histone demethylase KDM6A directly senses oxygen to control chromatin and cell fate. *Science* **363**, 1217-1222, doi:10.1126/science.aaw1026 (2019).
- 3 Semenza, G. L. Pharmacologic Targeting of Hypoxia-Inducible Factors. *Annu Rev Pharmacol Toxicol* **59**, 379-403, doi:10.1146/annurev-pharmtox-010818-021637 (2019).
- 4 Semenza, G. L. Oxygen sensing, hypoxia-inducible factors, and disease pathophysiology. *Annu Rev Pathol* **9**, 47-71, doi:10.1146/annurev-pathol-012513-104720 (2014).
- 5 Schito, L. & Semenza, G. L. Hypoxia-Inducible Factors: Master Regulators of Cancer Progression. *Trends Cancer* **2**, 758-770, doi:10.1016/j.trecan.2016.10.016 (2016).
- 6 Rankin, E. B., Nam, J. M. & Giaccia, A. J. Hypoxia: Signaling the Metastatic Cascade. *Trends Cancer* **2**, 295-304, doi:10.1016/j.trecan.2016.05.006 (2016).
- 7 Shen, C. *et al.* Genetic and functional studies implicate HIF1alpha as a 14q kidney cancer suppressor gene. *Cancer Discov* **1**, 222-235, doi:10.1158/2159-8290.CD-11-0098 (2011).
- 8 Mazumdar, J. *et al.* HIF-2alpha deletion promotes Kras-driven lung tumor development. *Proc Natl Acad Sci U S A* **107**, 14182-14187, doi:10.1073/pnas.1001296107 (2010).
- 9 Daniels, D. L. *et al.* Mutual Exclusivity of MED12/MED12L, MED13/13L, and CDK8/19 Paralogs Revealed within the CDK-Mediator Kinase Module. *Journal of Proteomics and Bioinformatics* **S2**, doi:10.4172/jpb.S2-004 (2013).
- 10 Galbraith, M. D. *et al.* HIF1A employs CDK8-mediator to stimulate RNAPII elongation in response to hypoxia. *Cell* **153**, 1327-1339, doi:10.1016/j.cell.2013.04.048 (2013).
- 11 Nemet, J., Jelacic, B., Rubelj, I. & Sopta, M. The two faces of Cdk8, a positive/negative regulator of transcription. *Biochimie* **97**, 22-27, doi:10.1016/j.biochi.2013.10.004 (2014).
- 12 Galbraith, M. D., Donner, A. J. & Espinosa, J. M. CDK8: A positive regulator of transcription. *Transcr* **1**, 4-12, doi:10.4161/trns.1.1.12373 (2010).
- 13 Steinparzer, I. *et al.* Transcriptional Responses to IFN-gamma Require Mediator Kinase-Dependent Pause Release and Mechanistically Distinct CDK8 and CDK19 Functions. *Mol. Cell* **76**, 485-499 e488, doi:10.1016/j.molcel.2019.07.034 (2019).
- 14 Knuesel, M. T., Meyer, K. D., Bernecky, C. & Taatjes, D. J. The human CDK8 subcomplex is a molecular switch that controls Mediator coactivator function. *Genes Dev* **23**, 439-451 (2009).
- 15 Audetat, K. A. *et al.* A Kinase-Independent Role for Cyclin-Dependent Kinase 19 in p53 Response. *Mol Cell Biol* **37**, doi:10.1128/MCB.00626-16 (2017).
- 16 Cee, V. J., Chen, D. Y., Lee, M. R. & Nicolaou, K. C. Cortistatin A is a high-affinity ligand of protein kinases ROCK, CDK8, and CDK11. *Angew Chem Int Ed Engl* **48**, 8952-8957, doi:10.1002/anie.200904778 (2009).
- 17 Rzymiski, T., Mikula, M., Wiklik, K. & Brzozka, K. CDK8 kinase--An emerging target in targeted cancer therapy. *Biochim Biophys Acta* **1854**, 1617-1629, doi:10.1016/j.bbapap.2015.05.011 (2015).
- 18 Bergeron, P. *et al.* Design and Development of a Series of Potent and Selective Type II Inhibitors of CDK8. *ACS Med Chem Lett* **7**, 595-600, doi:10.1021/acsmedchemlett.6b00044 (2016).
- 19 Czodrowski, P. *et al.* Structure-Based Optimization of Potent, Selective, and Orally Bioavailable CDK8 Inhibitors Discovered by High-Throughput Screening. *J Med Chem* **59**, 9337-9349, doi:10.1021/acs.jmedchem.6b00597 (2016).
- 20 Koehler, M. F. *et al.* Development of a Potent, Specific CDK8 Kinase Inhibitor Which Phenocopies CDK8/19 Knockout Cells. *ACS Med. Chem. Lett.* **7**, 223-228, doi:10.1021/acsmedchemlett.5b00278 (2016).
- 21 Mallinger, A. *et al.* 2,8-Disubstituted-1,6-Naphthyridines and 4,6-Disubstituted-Isoquinolines with Potent, Selective Affinity for CDK8/19. *ACS Med. Chem. Lett.* **7**, 573-578, doi:10.1021/acsmedchemlett.6b00022 (2016).

- 22 Mallinger, A. *et al.* Discovery of Potent, Selective, and Orally Bioavailable Small-Molecule Modulators of the Mediator Complex-Associated Kinases CDK8 and CDK19. *J Med Chem* **59**, 1078-1101, doi:10.1021/acs.jmedchem.5b01685 (2016).
- 23 Schiemann, K. *et al.* Discovery of potent and selective CDK8 inhibitors from an HSP90 pharmacophore. *Bioorg Med Chem Lett* **26**, 1443-1451, doi:10.1016/j.bmcl.2016.01.062 (2016).
- 24 Johannessen, L. *et al.* Small-molecule studies identify CDK8 as a regulator of IL-10 in myeloid cells. *Nature chemical biology* **13**, 1102-1108, doi:10.1038/nchembio.2458 (2017).
- 25 Hatcher, J. M. *et al.* Development of Highly Potent and Selective Steroidal Inhibitors and Degraders of CDK8. *ACS Med Chem Lett* **9**, 540-545, doi:10.1021/acsmedchemlett.8b00011 (2018).
- 26 Akoulitchev, S., Chuikov, S. & Reinberg, D. TFIID is negatively regulated by cdk8-containing mediator complexes. *Nature* **407**, 102-106, doi:10.1038/35024111 (2000).
- 27 Fryer, C. J., White, J. B. & Jones, K. A. Mastermind recruits CycC:CDK8 to phosphorylate the Notch ICD and coordinate activation with turnover. *Mol Cell* **16**, 509-520 (2004).
- 28 Pelish, H. E. *et al.* Mediator kinase inhibition further activates super-enhancer-associated genes in AML. *Nature* **526**, 273-276, doi:10.1038/nature14904 (2015).
- 29 Donner, A. J., Ebmeier, C. C., Taatjes, D. J. & Espinosa, J. M. CDK8 is a positive regulator of transcriptional elongation within the serum response network. *Nat Struct Mol Biol* **17**, 194-201, doi:10.1038/nsmb.1752 (2010).
- 30 Love, M. I., Huber, W. & Anders, S. Moderated estimation of fold change and dispersion for RNA-seq data with DESeq2. *Genome Biol.* **15**, 550, doi:10.1186/s13059-014-0550-8 (2014).
- 31 Galbraith, M. D. *et al.* CDK8 Kinase Activity Promotes Glycolysis. *Cell Rep* **21**, 1495-1506, doi:10.1016/j.celrep.2017.10.058 (2017).
- 32 Smythies, J. A. *et al.* Inherent DNA-binding specificities of the HIF-1alpha and HIF-2alpha transcription factors in chromatin. *EMBO Rep.* **20**, doi:10.15252/embr.201846401 (2019).
- 33 Schodel, J., Mole, D. R. & Ratcliffe, P. J. Pan-genomic binding of hypoxia-inducible transcription factors. *Biol Chem* **394**, 507-517, doi:10.1515/hsz-2012-0351 (2013).
- 34 Platt, J. L. *et al.* Capture-C reveals preformed chromatin interactions between HIF-binding sites and distant promoters. *EMBO Rep.* **17**, 1410-1421, doi:10.15252/embr.201642198 (2016).
- 35 Andrysik, Z. *et al.* Identification of a core TP53 transcriptional program with highly distributed tumor suppressive activity. *Genome Res.* **27**, 1645-1657, doi:10.1101/gr.220533.117 (2017).
- 36 Zaret, K. S. & Carroll, J. S. Pioneer transcription factors: establishing competence for gene expression. *Genes Dev* **25**, 2227-2241, doi:10.1101/gad.176826.111 (2011).
- 37 Liu, J. *et al.* An Integrated TCGA Pan-Cancer Clinical Data Resource to Drive High-Quality Survival Outcome Analytics. *Cell* **173**, 400-416 e411, doi:10.1016/j.cell.2018.02.052 (2018).

Reviewer #1 (Remarks to the Author):

The authors addressed a vast majority of my comments and the two parts of the revised manuscript is now well connected. The manuscript is also vastly improved in terms of an overall message and is now, in my opinion, suitable for publication, with minor overview/corrections. None of the comments are major, there are a couple.

Line 91. I do not see a difference in repression between wild type and Hif-/- cells. Repression actually appears to be exactly the same based on visuals of figure 1c. The numbers above that are different, but those are based on statistical thresholds within each dataset and due to their small changes are less believable than activation figures. One way to address this is to show values for all repressed genes in each of the datasets regardless of whether they are statistically repressed or not. Alternatively, one may tone down the statement in line 91 about repression, either remove repression or use a qualified such as "possibly".

117-122. I could not find information on how PRO-seq datasets were normalized, especially when promoter counts and downstream counts may have been used for DeSeq2 pipelines separately. I am not sure one can do that, especially for promoters, based purely on sequencing coverage. I am aware of two methods to normalize based on Lis and Danko labs's work, none being perfect, first, the overall sequencing depth sans ribosomal RNA or, secondly, based on ends of longest genes. If it was done by straight reads per million, the percentages of ribosomal RNA reads should be specified. They tend to vary and thus can make skew FPKM normalization. The former method may be the method of choice due to long-ish incubation times with hypoxia. Either way, how this was done should be specified in methods and I apologize again if I'd missed it.

137. It might be better to delete "but not downregulated" or replace with a more toned-down statement such as "compared to downregulated", since both groups of genes do show an increase and one can argue that it is pretty strong in each case when viewed on its own merits.

155-160. I am unclear about why bidirectional transcription is emphasized. There is nothing that is being made of it elsewhere in the manuscript and there is also no distinction between bidirectional transcription/promoters and unidirectional. It seems to me that transcription at these sites increases, period, and that transcription is bidirectional is a secondary observation. Unless I missed it, there is no data shown that hypoxia affects bidirectional genes preferentially or that it makes transcription on the same genes bidirectional. The fact that transcription is dampened in -/- cells does not appear to have anything to do with bidirectionality.

Lines 311 and 318: Maybe rephrase "the exercise"?

Reviewer #2 (Remarks to the Author):

The authors have thoroughly addressed all the points raised in the original manuscript. Overall, this work is of high-quality and I highly recommend publishing it.

Reviewer #3 (Remarks to the Author):

I have no additional comments or concerns.

Point-by-point response to Reviewer 1.

We are grateful for Reviewer 1's constructive feedback and the overall positive assessment of the revised manuscript. Our responses to the comments to the revised manuscript are as follows:

Line 91. I do not see a difference in repression between wild type and Hif^{-/-} cells. Repression actually appears to be exactly the same based on visuals of figure 1c. The numbers above that are different, but those are based on statistical thresholds within each dataset and due to their small changes are less believable than activation figures. One way to address this is to show values for all repressed genes in each of the datasets regardless of whether they are statistically repressed or not. Alternatively, one may tone down the statement in line 91 about repression, either remove repression or use a qualified such as “possibly”.

Response: We agree that the Volcano plot in Fig. 1c is mostly highlighting the differences in **p-values**, rather than expression, between the WT and HIF1A^{-/-} cells. However, in line with the suggestion by the Reviewer, Fig. 1d indeed compares the **fold changes** for all genes (regardless of statistical cut off, with those passing the cut off for repression in wild type cells being labelled in green)– with the majority of significantly repressed genes displaying a shift toward less repression during hypoxia in HIF1A^{-/-} cells (i.e. right of the diagonal). This point is further illustrated in the sina plots in Supplementary Figure 2a. Lastly, the metagene in Supplementary Fig. 2d displays the average transcriptional profile for this gene set, confirming lower reads across gene bodies in hypoxia only in HIF1A wild-type cells. Nevertheless, to tone down this sentence as suggested by the Reviewer, we removed the word ‘strongly’ from ‘with activation and repression **strongly** reduced in HIF1A^{-/-} cells’

117-122. I could not find information on how PRO-seq datasets were normalized, especially when promoter counts and downstream counts may have been used for DeSeq2 pipelines separately. I am not sure one can do that, especially for promoters, based purely on sequencing coverage. I am aware of two methods to normalize based on Lis and Danko labs's work, none being perfect, first, the overall sequencing depth sans ribosomal RNA or, secondly, based on ends of longest genes. If it was done by straight reads per million, the percentages of ribosomal RNA reads should be specified. They tend to vary and thus can make skew FPKM normalization. The former method may be the method of choice due to long-ish incubation times with hypoxia. Either way, how this was done should be specified in methods and I apologize again if I'd missed it.

Response: Thanks for this comment, which prompted us to provide more detail about our normalization method. Normalization was based on the number of reads within TSS and body regions of annotated genes with lengths greater than 1500 bp, a common practice that allows for separation into TSS and gene body regions of sufficient length. This also takes care of the rRNA issue, because all the rRNA genes in the RefSeq Hg19 annotation are less than 1500 bp and thus not considered in our analysis. Although we normalized separately for TSSs and body regions, the resulting relative values for TSS and gene bodies do not deviate significantly when compared to a combined normalization strategy. We have added further details to the Methods in lines 888-890.

137. It might be better to delete “but not downregulated” or replace with a more toned-down statement such as “compared to downregulated”, since both groups of genes do show an increase and one can argue that it is pretty strong in each case when viewed on its own merits.

Response: Thanks for this comment, which prompted a revision of this fragment. In the revised text, we clarify that although ChIP-seq enrichment signal increases at TSSs at all classes of genes upon hypoxia, only those that are acutely upregulated display stronger binding around their TSS during hypoxia relative to genes with non-significant (n.s.) differences in transcription.

155-160. I am unclear about why bidirectional transcription is emphasized. There is nothing that is being made of it elsewhere in the manuscript and there is also no distinction between bidirectional transcription/promoters and unidirectional. It seems to me that transcription at these sites increases, period, and that transcription is bidirectional is a secondary observation. Unless I missed it, there is no data shown that hypoxia affects bidirectional genes preferentially or that it makes transcription on the same genes bidirectional. The fact that transcription is dampened in -/- cells does not appear to have anything to do with bidirectionality.

Response: Thanks for the comment, which prompted further clarification in the revised text. The emphasis on bidirectional transcription while characterizing HIF1A enhancers arises from previous work showing that bidirectional intergenic transcription is a hallmark of active enhancer regions¹⁻³. We do not mean to suggest that bidirectionality specifically is affected by hypoxia or HIF1A, or that is exclusive to enhancers and not promoters. The revised text clarifies this issue.

Lines 311 and 318: Maybe rephrase “the exercise”?

Response: Thanks, we have reworded these two sentences.

References

- 1 Andersson, R. *et al.* An atlas of active enhancers across human cell types and tissues. *Nature* **507**, 455-461, doi:10.1038/nature12787 (2014).
- 2 Danko, C. G. *et al.* Signaling pathways differentially affect RNA polymerase II initiation, pausing, and elongation rate in cells. *Mol Cell* **50**, 212-222, doi:10.1016/j.molcel.2013.02.015 (2013).
- 3 Kim, T. K. *et al.* Widespread transcription at neuronal activity-regulated enhancers. *Nature* **465**, 182-187, doi:10.1038/nature09033 (2010).